# Hypertrophic cardiomyopathy mutations in the pliant and light chain-binding regions of the lever arm of human β-cardiac myosin have divergent effects on myosin function

**Makenna M Morck[1,2], Debanjan Bhowmik[1,2†], Divya Pathak[1,2], Aminah Dawood[1,2], James Spudich[1,2], Kathleen M Ruppel[1,2]\***

[1]Stanford Cardiovascular Institute, Stanford University School of Medicine, Stanford, United States; [2]Department of Biochemistry, Stanford University School of Medicine, Stanford, United States

**\*For correspondence:**
kruppel@stanford.edu

**Present address:**
[†]Transdisciplinary Research Program, Rajiv Gandhi Centre for Biotechnology, Thiruvananthapuram, India

**Abstract** Mutations in the lever arm of β-cardiac myosin are a frequent cause of hypertrophic cardiomyopathy, a disease characterized by hypercontractility and eventual hypertrophy of the left ventricle. Here, we studied five such mutations: three in the pliant region of the lever arm (D778V, L781P, and S782N) and two in the light chain-binding region (A797T and F834L). We investigated their effects on both motor function and myosin subfragment 2 (S2) tail-based autoinhibition. The pliant region mutations had varying effects on the motor function of a myosin construct lacking the S2 tail: overall, D778V increased power output, L781P reduced power output, and S782N had little effect on power output, while all three reduced the external force sensitivity of the actin detachment rate. With a myosin containing the motor domain and the proximal S2 tail, the pliant region mutations also attenuated autoinhibition in the presence of filamentous actin but had no impact in the absence of actin. By contrast, the light chain-binding region mutations had little effect on motor activity but produced marked reductions in autoinhibition in both the presence and absence of actin. Thus, mutations in the lever arm of β-cardiac myosin have divergent allosteric effects on myosin function, depending on whether they are in the pliant or light chain-binding regions.

## Editor's evaluation

This paper explores several mutations lying in the lever arm region of cardiac myosin. Using a diverse array of biochemical, kinetic and biophysical techniques the authors show that the individual mutations can have many effects, not only on the kinetic properties of the ATPase cycle of the myosin, but also on the ability of the myosin to adopt an auto-inhibited conformation involving regions of the myosin outside of the motor domain. It shows the value of examining disease causing mutations of myosin using a variety of techniques.

## Introduction

The myosin lever arm was first recognized almost 30 years ago as the first structure of myosin subfragment 1 (S1) was described (*Rayment et al., 1993a*; *Rayment et al., 1993b*), and the lever arm's function in amplifying the motion of the converter domain was subsequently confirmed (*Uyeda et al., 1996*; *Geeves and Holmes, 1999*). Much work since has focused on understanding the functional

significance of different lever arm features across myosin classes (*Spudich and Sivaramakrishnan, 2010*). However, less attention has been paid to the functional consequences of point mutations in the myosin lever arm. Such mutations in the human β-cardiac myosin lever arm are a frequent cause of hypertrophic cardiomyopathy (HCM), a disease characterized by hypercontractility and eventual hypertrophy of the left ventricle (*Ho et al., 2009*). While many HCM-causing mutations in the myosin motor domain have been described and characterized (*Nag et al., 2015*; *Adhikari et al., 2016*; *Kawana et al., 2017*; *Nag et al., 2017*; *Adhikari et al., 2019*; *Sarkar et al., 2020*; *Vander Roest et al., 2021*; *Spudich, 2019*; *Liu et al., 2018*), HCM-causing mutations in the lever arm remain understudied, despite the fact that the lever arm has a relatively high rate of these mutations and plays a key role in transducing the chemical energy of ATP hydrolysis into physical motion (*Uyeda et al., 1996*; *Geeves and Holmes, 1999*).

HCM is a leading cause of genetic heart disease, affecting up to 1 in 200 in the US population (*Semsarian et al., 2015*). Mutations leading to HCM have been found in genes encoding a variety of sarcomeric proteins; however, the vast majority occur in either the β-cardiac myosin heavy chain or cardiac myosin-binding protein-C (*Konno et al., 2010*). Because left ventricular hypercontractility typically precedes hypertrophy in HCM patients (*Ho et al., 2009*), it has been hypothesized that sarcomeric mutations lead to hypercontractility at the molecular scale. Early work in this area suggested that mutations might increase myosin's motor activity by increasing its actin-activated ATPase rate, motility velocity, or force production (*Tyska et al., 2000*; *Debold et al., 2007*). While some mutations may function by this mechanism (*Adhikari et al., 2016*; *Adhikari et al., 2019*; *Liu et al., 2018*), lately, a more compelling hypothesis has gained traction: many HCM-causing mutations appear to reduce myosin's ability to form an autoinhibited state (*Nag et al., 2017*; *Adhikari et al., 2019*; *Sarkar et al., 2020*; *Vander Roest et al., 2021*; *Spudich, 2019*; *Moore et al., 2012*; *Spudich, 2015*; *Alamo et al., 2017*; *Robert-Paganin et al., 2018*; *Anderson et al., 2018*). Loss of an autoinhibited state could lead to additional myosin heads acting to generate force during contraction, thus leading to hypercontractility.

An autoinhibited state of myosin was first described in smooth muscle myosin over 20 years ago (*Trybus, 1991*), but its relevance to β-cardiac myosin and HCM was only more recently recognized. In the structural view of this smooth muscle autoinhibited state, the myosin heads fold back onto their own subfragment 2 (S2) tail in a conformation known as the interacting heads motif (IHM; *Wendt et al., 2001*; *Woodhead et al., 2005*). One of the two heads in the dimer has its actin-binding interface buried in the folded structure; this head is referred to as the 'blocked head', while the other is called the 'free head', since its actin-binding interface is not hidden structurally. Many myosin types have been shown to assume this folded back IHM structure (*Woodhead et al., 2005*; *Jung et al., 2008*; *Al-Khayat et al., 2013*), and it is now thought to be a common feature across the myosin II class (*Lee et al., 2018*).

The IHM structure has been correlated to an ultra-low basal ATPase rate (three orders of magnitude below the actin-activated ATPase rate) in the absence of actin called the 'super relaxed state' (SRX; *Anderson et al., 2018*; *Al-Khayat et al., 2013*; *Stewart et al., 2010*; *McNamara et al., 2015*; *Rohde et al., 2018*). Heads lacking the S2 tail mostly have a faster basal ATPase rate (two orders of magnitude below the actin-activated ATPase rate) referred to as the 'disordered relaxed state' (DRX). Additionally, actin-activated ATPases comparing myosin with and without its S2 tail have shown that when the S2 tail is present, the apparent ATPase rate decreases, suggesting that the S2 tail promotes autoinhibition (*Nag et al., 2017*; *Adhikari et al., 2019*; *Sarkar et al., 2020*; *Vander Roest et al., 2021*; *Trybus et al., 1997*). Therefore, myosin containing the S2 tail is thought to be in equilibrium between an open state available for actin binding and the closed IHM conformation. While these structural states are correlated to specific basal and actin-activated ATPase rates, there may be circumstances in which the structural states are uncoupled from their respective rates (*Chu et al., 2021*; *Nag and Trivedi, 2021*). Thus, while these functional assays measure autoinhibition by the myosin S2 tail, they are only correlative measures for structural states (i.e., the IHM).

HCM-causing mutations in the myosin lever arm could lead to hypercontractility by (1) disrupting S2 tail-based autoinhibition, (2) increasing intrinsic motor activity without affecting autoinhibition, or (3) affecting both intrinsic motor activity and autoinhibition. The lever arm's role in the formation of the autoinhibited state has previously been explored in mollusk myosin, which is primarily regulated via molecular switching from the autoinhibited off state to an open state in the presence of $Ca^{2+}$. A

number of structural studies collectively concluded that three joints in the molluscan myosin lever arm appear to be potential sources of flexibility which may be necessary for the formation of the IHM (*Houdusse et al., 2000*; *Houdusse and Cohen, 1996*; *Himmel et al., 2009*; *Pylypenko and Houdusse, 2011*; *Brown et al., 2011*): a region at the extreme N-terminus of the lever arm, termed as the 'pliant' region, a typical bent region between the essential light chain (ELC) and the regulatory light chain (RLC) binding regions, and a region at the C-terminus of the lever arm in the RLC-binding area termed the 'hook' or 'ankle' joint. Recently, high-resolution cryo-electron microscopy (cryo-EM) structures of the folded-back IHM state in smooth muscle myosin demonstrated that the lever arm must take on a different conformation in each of the asymmetric folded heads in the IHM, further confirming the importance of lever arm positioning in the folded state (*Scarff et al., 2020*; *Yang et al., 2020*; *Heissler et al., 2021*). Mutations in the lever arm could reduce myosin's S2 tail-based autoinhibition by negatively impacting any of these structural requirements for forming the IHM. Alternatively, lever arm mutations could increase intrinsic motor activity, leading to hypercontractility. For example, mutations could increase the rigidity of the lever arm and/or alter its light chain-binding properties, which could in turn influence power output. Indeed, several previous studies investigating RLC mutations have suggested that mutations can modulate lever arm compliance (*Greenberg et al., 2010*; *Sherwood et al., 2004*; *Karabina et al., 2015*; *Farman et al., 2014*). Lever arm mutations could also allosterically affect the motor domain, potentially leading to increased motor activity.

In the present study, we examined five adult-onset HCM-causing mutations in the myosin lever arm to determine how they affect myosin function: D778V, L781P, S782N, A797T, and F834L (*Figure 1*). We selected these mutations based on their confirmed pathogenicity and to allow investigation of mutations across the lever arm. We also aimed to investigate mutations in or near regions thought to be important in lever arm function, including the pliant region (D778V, L781P, and S782N), the bent region between the light chains (A797T), and the hook joint (F834L). We hypothesized that mutations in the lever arm could affect myosin either by (1) altering its intrinsic motor activity, and/or (2) reducing its ability to form the autoinhibited state. We found that the three mutations in the pliant region of the lever arm, D778V, L781P, and S782N, led to changes in both myosin's motor activity and the formation of the autoinhibited state. The two mutations in the light chain-binding regions, A797T and F834L, clearly impacted myosin's ability to form the autoinhibited state, likely explaining their contributions to hypercontractility. Thus, mutations in the lever arm have varying impacts on myosin function depending on whether they are in the pliant region or the light chain-binding regions.

## Results

### Mutations do not significantly impact light chain loading

We first sought to determine whether any of the five mutations impacted the expected 1:1:1 stoichiometry of myosin heavy chain:ELC:RLC of our purified proteins. To test this, we used a Coomassie SDS-PAGE gel-based assay that quantified the ratio of myosin heavy chain to ELC and RLC (*Figure 2A, B*). For this assay, we used a two-headed myosin construct that contains the myosin head, full lever arm, and the first two heptads of the S2 tail, followed by a GCN4 leucine zipper moiety to ensure dimerization, a GFP tag, and a C-terminal affinity clamp peptide tag (2-hep myosin [*Nag et al., 2017*]). When compared to the paired wild-type (WT) 2-hep control, none of the mutants had significantly changed ELC loading (p vs WT for D778V = 0.085, L781*P*=0.17, S782N=0.53, A797T=0.78, F834L=0.95) or RLC loading (p vs WT for D778V = 0.14, L781*P*=0.28, S782N=0.32, A797T=0.82, F834L=0.78), suggesting that at least for these five mutations in the context of our purified system, light chain loading is not significantly impacted (*Figure 2C*).

### Only D778V impacts the actin-activated ATPase rate of two-headed myosin with a short tail

We then examined the effects of each of the five mutations on myosin's actin-activated ATPase activity of 2-hep β-cardiac myosin. We have previously shown that 2-hep myosin produces similar ATPase rates to S1 constructs, albeit with a slightly tighter apparent affinity for actin (K$_{app}$; *Adhikari et al., 2019*). Of the five mutations, only D778V produced a significantly different actin-activated ATPase rate as compared to WT, increasing the turnover number $k_{cat}$ by 16 ± 9% (p=0.041, average of three technical replicates of each of two biological replicates, compared to WT controls; *Figure 3A*; see raw

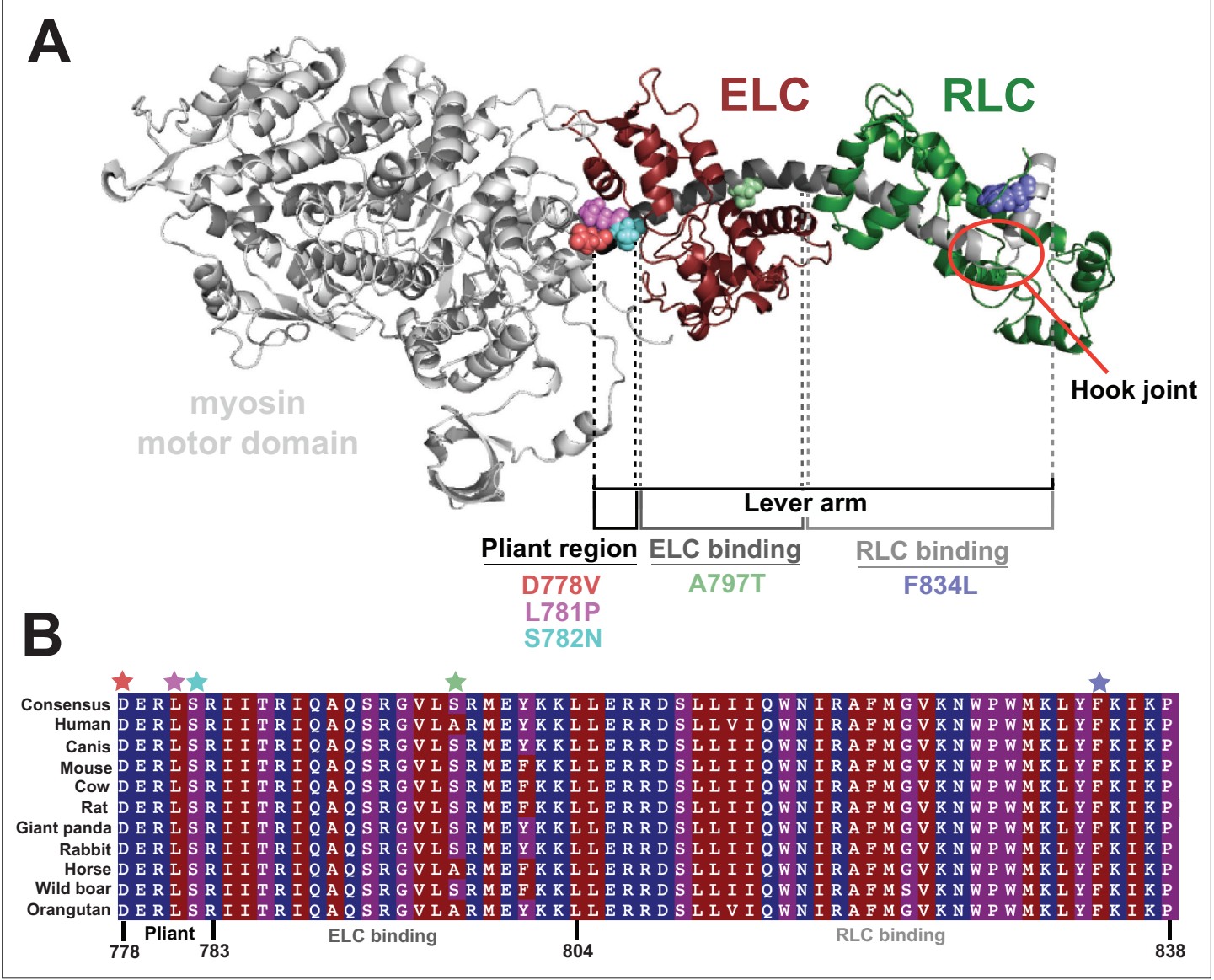

**Figure 1.** Location of hypertrophic cardiomyopathy-causing mutations along the lever arm. (**A**) Homology model of the prestroke β-cardiac myosin subfragment 1 (S1; *Homburger et al., 2016*) highlighting the five mutations examined in the present study: D778V, L781P, S782N, A797T, and F834L and their locations along the lever arm. The essential light chain (ELC) and regulatory light chain (RLC) are shown in maroon and green, respectively, and the hook joint is highlighted in the red circle. (**B**) Alignment of β-cardiac myosin lever arms for several model organisms, demonstrating the high degree of conservation of the lever arm across species. Residues are colored by hydrophobicity, where red is most hydrophobic, and blue is most hydrophilic. Mutated residues studied here are indicated with colored stars.

values in *Supplementary file 1* and raw plots in *Figure 3—figure supplement 1*). In contrast, L781P, S782N, A797T, and F834L all had actin-activated ATPase curves that very closely resembled their matched WT controls (*Figure 3B–E*; see raw values in *Supplementary file 1*).

## Lever arm mutations have varying effects on actin gliding velocity

We next assessed the effects of lever arm mutations on actin gliding velocity in an in vitro actin gliding motility assay (*Aksel et al., 2015*). For these experiments, we again used the 2-hep myosin construct, attached to the surface via interaction of its C-terminal affinity clamp to SNAP-PDZ. In the pliant region, the D778V mutation increased actin gliding velocity by 46 ± 8% (mean ± SD, p<0.0001), the L781P mutation reduced velocity by 30 ± 8% (mean ± SD, p<0.0001), and the S782N mutation had no significant effect on velocity (p=0.19) as compared to actin gliding velocities of same day WT 2-hep

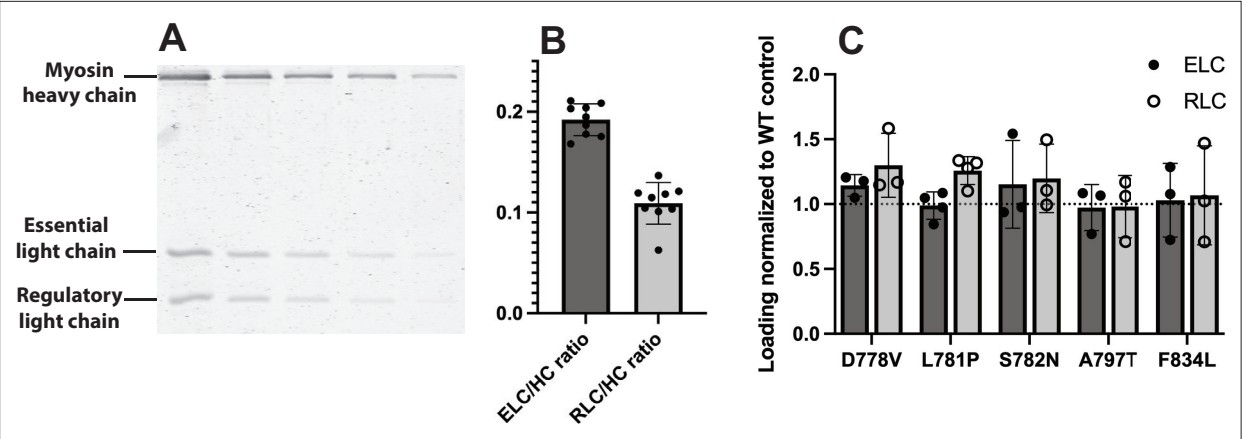

**Figure 2.** Light chain loading of wild type (WT) vs mutant myosins. (**A**) Representative gel used to quantitate light chain loading (L781P 2-hep is shown). For each sample, myosin was titrated in a denaturing SDS-PAGE gel, stained with Coomassie, and scanned for 700 nm fluorescence. See Materials and Methods. (**B**) Light chain loading across all WT 2-hep samples measured. Mean essential light chain/heavy chain (ELC/HC) = 0.19 ± 0.02, regulatory light chain/heavy chain (RLC/HC) = 0.11 ± 0.02 (mean ± SD). Expected ELC/HC and RLC/HC ratios, if Coomassie staining were unbiased, are 0.168 and 0.146. (**C**) Light chain loading of mutant 2-hep myosins normalized to WT controls. For each mutant, light chain loading was not statistically different from WT.

The online version of this article includes the following source data for figure 2:

**Source data 1.** Raw uncropped gel image.

**Source data 2.** Ratio of ELC and RLC to myosin 2-hep heavy chain.

controls (*Figure 3F*). In contrast, mutations in the light chain-binding domains had little to no effect on gliding velocities, as the A797T mutation reduced velocity by only 5 ± 6% (mean ± SD, p=0.033) and the F834L mutation had no significant effect on velocity (p=0.72; *Figure 3F*). Actin gliding velocity is primarily a function of detachment kinetics and step size (*Liu et al., 2018*). Thus, it appears that as a whole, mutations in the pliant region may have more impact on myosin's motor properties (i.e., attachment and detachment kinetics and step size), while the mutations in the light chain-binding regions may impact myosin's activity primarily through a different mechanism. Accordingly, we next sought to further dissect the effects of pliant region mutations using harmonic force spectroscopy (HFS).

## Harmonic force spectroscopy reveals altered detachment kinetics and step sizes due to pliant region mutations

We have previously described an optical trap setup called HFS that allows for the measurement of myosin's detachment rate from actin as a function of external load force (*Liu et al., 2018*; *Sung et al., 2015*; *Sung et al., 2017*) as well as its step size (*Vander Roest et al., 2021*). Myosin's detachment rate is reduced in the presence of resistive load forces and increased in the presence of assistive load forces, in accordance with the force-velocity relationship of contracting heart muscle (*Sung et al., 2015*; *Greenberg et al., 2014*). Here, we used HFS to measure the load-dependent detachment rates and step size of a short S1 (sS1) myosin construct containing the motor domain of myosin plus the first half of the lever arm (amino acids 1–808), allowing for the binding of the ELC but not the RLC, followed by a GFP tag.

We measured detachment kinetics of several molecules each of WT, D778V, L781P, and S782N and fit their behavior to the Arrhenius equation with a harmonic force correction as described previously (*Liu et al., 2018*; *Sung et al., 2015*; *Sung et al., 2017*), where $k_B$ is the Boltzmann constant and $T$ is temperature:

$$k_{det}\ (F, \Delta F) =\ k_0 I_0 \left( \frac{\Delta F \delta}{k_B T} \right) exp \left( \frac{-F \delta}{k_B T} \right) \tag{1}$$

This equation is a function of $k_0$, the detachment rate at zero load force, $\delta$, a measure of the force sensitivity, and $F$, the average external load force applied. Since the molecule experiences sinusoidal force, the equation is corrected with a zero-order Bessel function $I_0$ which is a function of $\Delta F$, the amplitude of the sinusoidal force. We then averaged the individual molecules (WT n=10

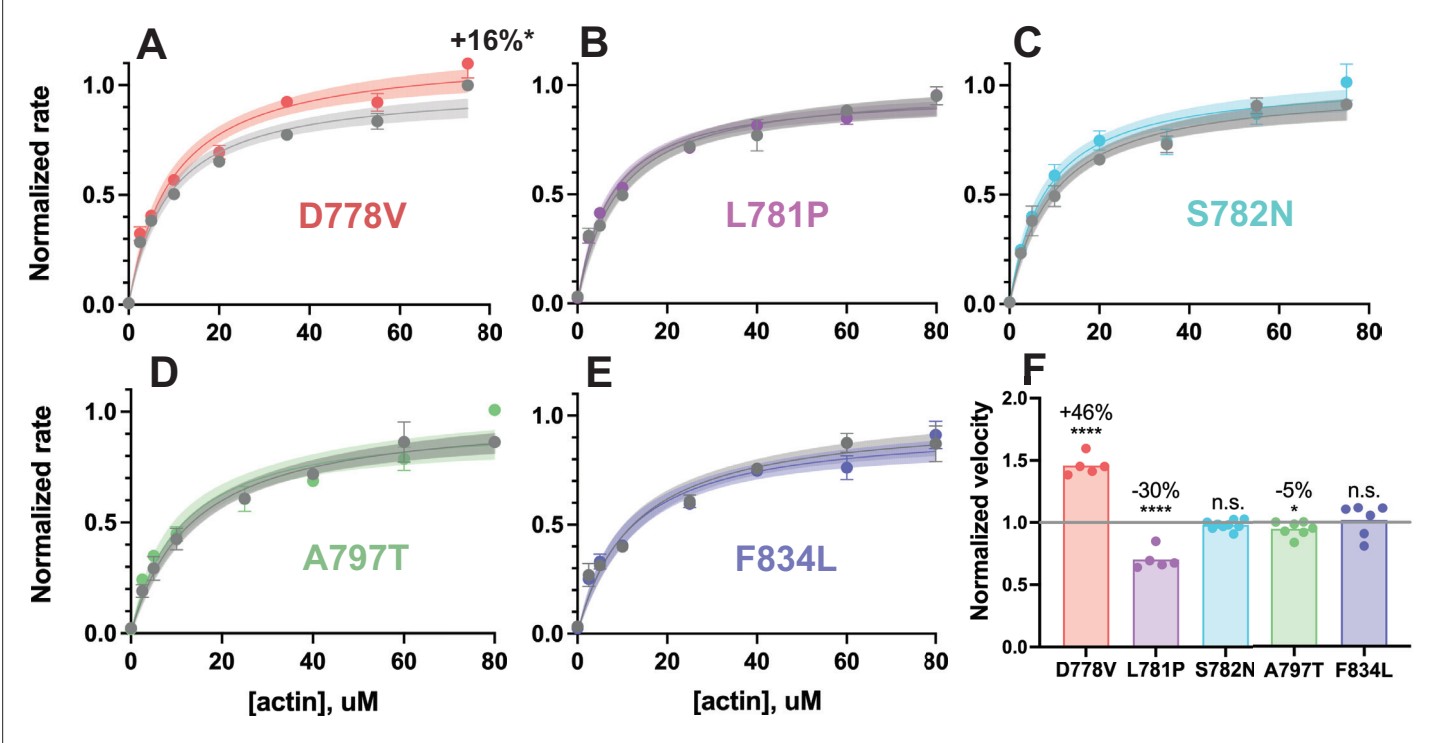

**Figure 3.** Actin-activated ATPase activity and in vitro motility velocity of 2-hep human β-cardiac myosin lever arm mutants vs wild-type (WT). (**A–E**) Representative actin-activated ATPase curves of 2-hep constructs for each lever arm mutant, normalized to their WT controls. Each plot shows a single biological replicate (one of two, see **Supplementary file 1**), where error bars represent the SD of three technical replicates. Where error bars are not shown, error is smaller than the size of the data point. Mutant data are plotted against their prep-matched WT 2-hep control (gray) and normalized to the WT $k_{cat}$ value. Curves are fit to Michaelis-Menten kinetics, and the shaded areas represent 95% CI of the fits. Only D778V produced a significant difference: the $k_{cat}$ was increased by 16 ± 9% (average of two independent biological replicates compared to WT controls). (**F**) In vitro motility velocities of lever arm mutants. Each data point represents the average velocity of the mutant 2-hep on a single slide normalized to the WT 2-hep velocity from the same slide. Bars represent the average of the data points. D778V increased the velocity by 46 ± 8% over WT, L781P reduced the velocity by 30 ± 8% compared to WT, A797T reduced the velocity by 5 ± 6%, and S782N and F834L had no significant effects on motility velocity (mean ± SD of the data points shown). **** indicates p≤0.0001, * indicates p≤0.05.

The online version of this article includes the following source data and figure supplement(s) for figure 3:

**Source data 1.** Rate vs (actin) for WT and mutant 2-hep myosins.

**Figure supplement 1.** Full actin-activated ATPase results.

**Figure supplement 1—source data 1.** Turnover rate per second vs (actin) of the actin-activated ATPase activity of each mutant 2-hep myosin and each mutant 25-hep myosin along with its same day WT 2-hep control.

molecules, D778V n=12, L781P n=22, and S782N n=11) to obtain the characteristic $k_0$ and $\delta$ values of WT and each lever arm mutant myosin. Interestingly, each pliant region mutation resulted in significant changes to the load force vs. detachment rate curve (**Figure 4A** and **Figure 4—figure supplement 1**), consistent with previous studies showing that mutations frequently impact force-dependent kinetics (**Vander Roest et al., 2021**; **Liu et al., 2018**). The fitted values for $k_0$ (WT = 147.0 ± 6.8 s⁻¹ (mean ± SEM); D778V = 315.6 ± 14.0 s⁻¹, p<0.0001; L781P = 101.7 ± 5.8 s⁻¹, p<0.0001; and S782N = 145.3 ± 6.7 s⁻¹, p=0.88; **Figure 4B**) roughly correlate to the changes we observed in motility velocity, consistent with the fact that motility velocity is detachment rate-limited (**Liu et al., 2018**). Interestingly, fitted values for $\delta$ demonstrate that all three pliant region mutations reduce force sensitivity (WT = 1.04 ± 0.05 nm (mean ± SEM); D778V = 0.68 ± 0.05 nm, p<0.0001; L781P = 0.81 ± 0.03 nm, p=0.0006; S782N = 0.83 ± 0.05 nm, p=0.0039; **Figure 4C**). We also measured the step size d of each mutation, and notably only L781P had any measurable impact on step size (WT = 4.3 ± 1.4 nm (mean ± SD); D778V = 4.5 ± 1.4 nm, p=0.66; L781P = 2.5 ± 1.4 nm, p=0.0032; S782N = 4.3 ± 1.2 nm, p=0.97; **Figure 4D**).

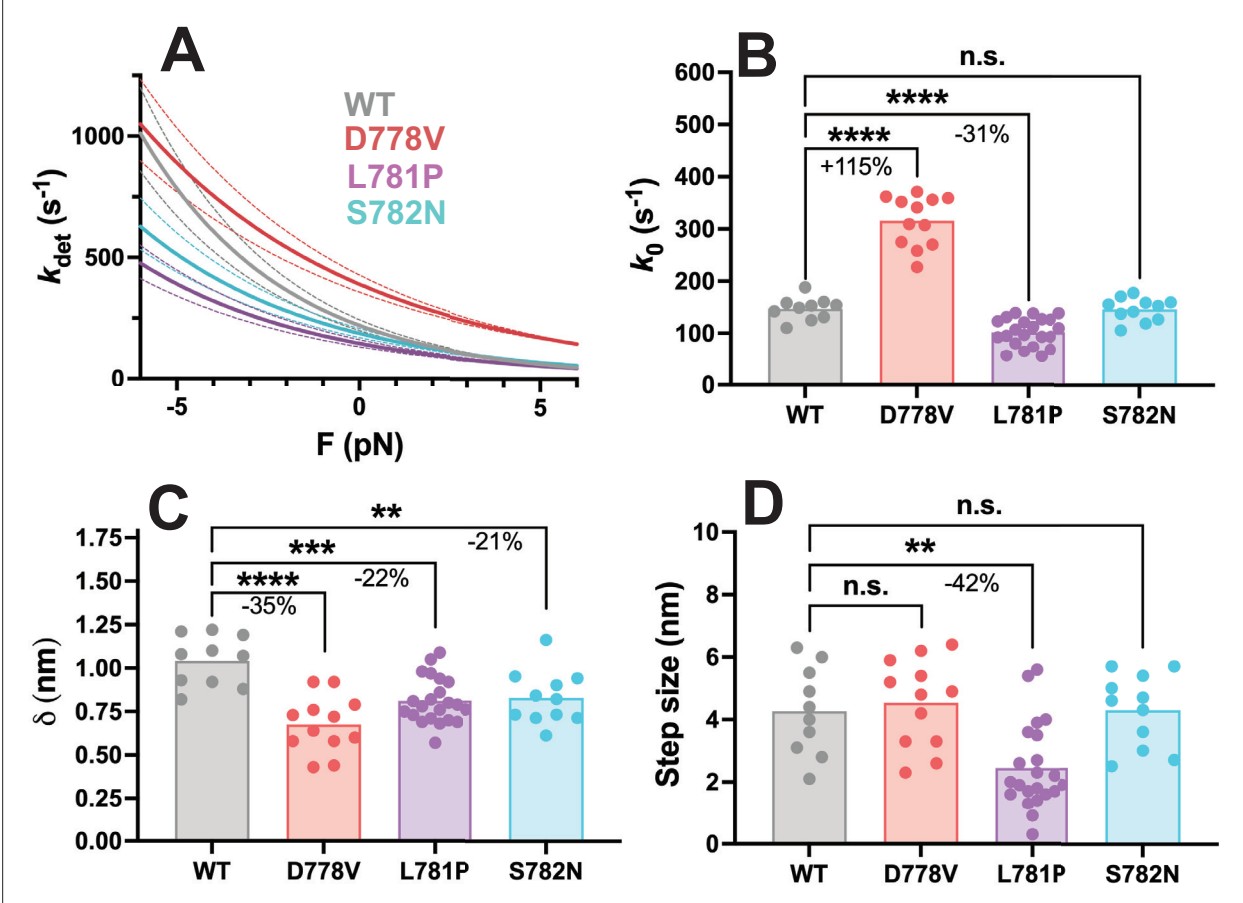

**Figure 4.** Harmonic force spectroscopy measurements of pliant region mutations in short subfragment 1 (sS1) human β-cardiac myosin. (**A**) Detachment rate of sS1 myosin as a function of load force, average of all molecules (wild type [WT]: n=10 molecules, D778V: n=12 molecules, L781P: n=22 molecules, S782N: n=11 molecules). WT curve is shown in gray, D778V curve is shown in red, L781P curve is shown in purple, and S782N curve is shown in cyan. Dotted lines show error propagated from SEM of fitted parameters. (**B**) $k_0$, the detachment rate at zero load force, for WT and each mutant sS1. D778V increased the detachment rate by 115 ± 9% (mean ± SEM), while L781P reduced the detachment rate by 31 ± 10%, and S782N had no significant effect on detachment rate. (**C**) $\delta$, the measure of force sensitivity, for WT and each mutant sS1. All the pliant region mutations reduced the force sensitivity: D778V by 35 ± 11%, L781P by 22 ± 8%, and S782N by 21 ± 10% (mean ± SEM). (**D**) Step size for WT and each mutation. Only L781P affected step size, reducing it by 42 ± 22% (mean ± SEM). Each data point represents the average value for one molecule, and bars show the average of the data points. ** indicates p≤0.01, *** indicates p≤0.001, **** indicates p≤0.0001.

The online version of this article includes the following source data and figure supplement(s) for figure 4:

**Source data 1.** Detachment rate, $\delta$, and step size for all molecules of WT and pliant region mutant myosins.

**Figure supplement 1.** Representative raw data and analysis for harmonic force spectroscopyHFS measurements.

**Figure supplement 1—source data 1.** (**I–L**) Data fof representative fitting using Arrhenius equation (corrected for harmonic force, *equation 1*) of the detachment rates ($K_{det}$) obtained at different load forces for one molecule each of WT (**I**), D778V (**J**), L781P (**K**), and S782N (**L**).

**Figure supplement 2.** Harmonic force spectroscopy measurements across unique protein preparations.

**Figure supplement 2—source data 1.** Detachment rate, delta, and step size for all molecules of WT and pliant region mutant myosins across separate myosin preps.

## Pliant region mutations have varied impacts on duty ratio, ensemble force, and power output

Given measured values for $k_{cat}$, $k_0$, step size, and the load-dependent detachment rate $k_{det}$, duty ratio $r$ as a function of load force $F$ can be calculated as follows *Liu et al., 2018*:

$$r\left(F\right) = \frac{k_{attach}}{k_{attach}+k_{det}\left(F\right)} \tag{2}$$

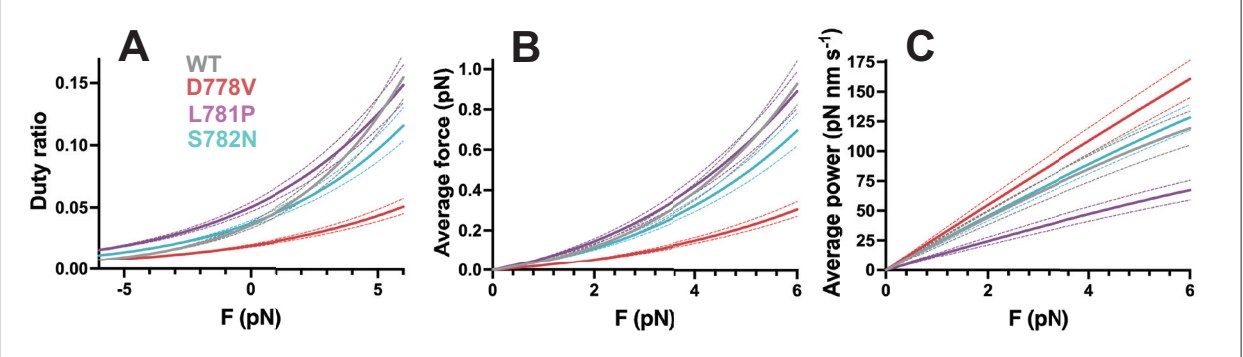

**Figure 5.** Effects of pliant region mutations on ensemble duty ratio, average force, and average power. Calculation of (**A**) duty ratio, (**B**) average force, and (**C**) average power output at resistive forces based on measured detachment rate $k_0$, actin-activated ATPase rate $k_{cat}$, force-dependent detachment rate $k_{det}(F)$, and step size (see *equations 2-6*). Wild type (WT) curves are shown in gray, D778V curves are shown in red, L781P curves are shown in purple, and S782N curves are shown in cyan, where dotted lines in the same color show error, propagated from the SEM of the individual molecules.

where $k_{attach}$ is the force-independent attachment rate, described by:

$$k_{attach} = \left( \frac{1}{k_{cat}} - \frac{1}{k_0} \right)^{-1} \tag{3}$$

The resulting force-dependent duty ratio is plotted in *Figure 5A*. Here, we plot only resistive forces since the ensemble behavior of heart muscle is to move against a load (blood pressure) during contraction. Duty ratio is increased when myosin spends more of its catalytic cycle in the force-producing state, bound to actin. Thus, mutant motors with faster detachment rates have lower duty ratios. In this case, the effect of reduced $\delta$ in pliant mutations had the strongest effect on duty ratio: even for S782N, which *only* affected force sensitivity, duty ratio was lowered at resistive forces. For D778V, this effect was compounded by the marked increase in $k_0$, which was not offset by the relatively smaller increase in $k_{cat}$. For L781P, the effect of reduced $\delta$ dominated the decreased $k_0$, except at high resistive force.

Using duty ratio, we can also calculate the average force that an individual myosin would exert in an ensemble system, which is again dependent on the load force (*Liu et al., 2018*):

$$F_{av}(F) = F\, r(F) \tag{4}$$

Results from this calculation are plotted in *Figure 5B*. The overall trends for average force resemble that of duty ratio because average force is just load force (the x-axis) multiplied by duty ratio.

Finally, power output is a function of force × velocity. Velocity itself a function of detachment rate $k_{det}$ and step size $d$:

$$vel(F) = k_{det}(F) \times d \tag{5}$$

such that power output is calculated by:

$$P_{av}(F) = vel(F) \times F_{av}(F) \tag{6}$$

Results from this calculation are plotted in *Figure 5C*. In this case, the increased velocity of D778V myosin dominates over the reduced average force, resulting in an increase in power output, particularly at high resistive forces. For L781P, the reduction in both step size and detachment rate results in a decrease in power output. S782N, by contrast, is relatively similar to WT in its power output.

## Lever arm mutants result in increased actin-activated ATPase when the S2 tail is present

To assess whether lever arm mutations impact myosin S2 tail-based autoinhibition, we introduced the mutations into a 25-hep myosin construct that is identical to 2-hep myosin, except that it contains the first 25 heptads of the S2 tail, rather than only the first 2 heptads. Thus, the myosin tail is long enough in the 25-hep construct for the myosin heads to fold back onto the tail. We have previously shown

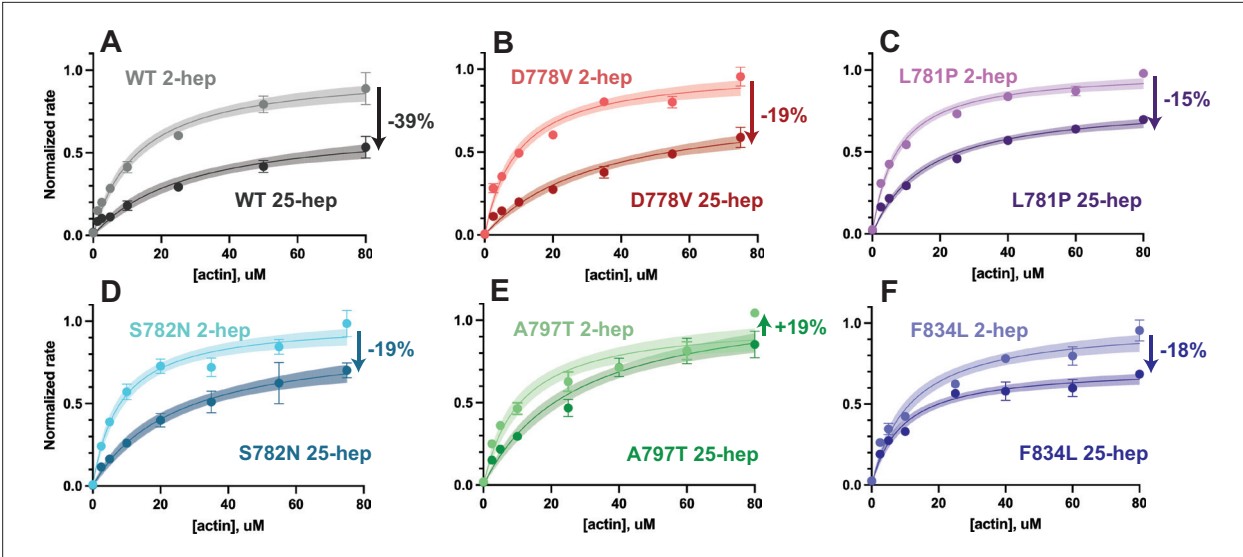

**Figure 6.** Actin-activated ATPase rates of 2-hep vs 25-hep β-cardiac myosin constructs. (**A–F**) Representative actin-activated ATPase curves for each mutant, normalized to the mutant 2-hep control $k_{cat}$. (**A**) is reproduced from *Vander Roest et al., 2021*. Each data point represents the average of three technical replicates of one biological replicate (one of two, see *Figure 3—figure supplement 1* and *Supplementary file 1*), and error bars represent SDs. Where error bars are not shown, error is smaller than the size of the data point. Curves are fitted to Michaelis-Menten kinetics, and shaded areas represent the 95% CI of the fits. Arrows with percentages on each graph show the percent change from the 2-hep to the matched 25-hep (average of both biological replicates, not just the representative curve shown). See *Supplementary file 1* for full results.

The online version of this article includes the following source data for figure 6:

**Source data 1.** Rate vs (actin) for wild type (WT) and mutant 2- vs 25-hep myosin.

that this 25-hep myosin has a reduced $k_{cat}$ as compared to the 2-hep myosin, such that a significant proportion of the 25-hep myosin is autoinhibited by the presence of the S2 tail (*Nag et al., 2017*; *Trybus et al., 1997*). This autoinhibition could arise from the IHM structure, where some of the myosin heads are sterically unable to bind actin. Mutations that disrupt autoinhibition show smaller differences between the 2-hep and 25-hep $k_{cat}$'s (*Adhikari et al., 2019*; *Sarkar et al., 2020*; *Vander Roest et al., 2021*).

Here, we found that all five of the lever arm mutations showed smaller differences between 2-hep and 25-hep as compared to WT (*Figure 6*, see full values in *Supplementary file 1* and raw plots in *Figure 3—figure supplement 1*). The mutations all resulted in smaller differences between 2- and 25-hep myosin (WT 2-hep/25-hep ratio = 0.61 ± 0.10; D778V = 0.81 ± 0.15, p vs. WT ratio = 0.0116; L781*P* = 0.85 ± 0.10, p=0.007; S782N = 0.81 ± 0.12, p=0.033; A797T = 1.19 ± 0.16, p=0.0001; F834L = 0.82 ± 0.10; p=0.0281), with reductions comparable to some previous mutations we have measured (D382Y [*Adhikari et al., 2019*], R403Q, and R663H [*Sarkar et al., 2020*]).

## Mutations in the light chain-binding domain reduce SRX, while pliant region mutations do not reduce SRX

To determine the effect of lever arm mutations on the SRX, we next measured the single-turnover ATPase activity of the myosins in the absence of actin. In this experiment, myosin is loaded with a roughly equimolar amount of fluorescent mant-ATP, then chased with excess unlabeled 'dark' ATP (*Anderson et al., 2018*; *Stewart et al., 2010*; *Rohde et al., 2018*). The fluorescence of the sample decays, and the decay rate can only be adequately fit with a double-exponential function (it does not fit a single exponential), suggesting the presence of two distinct structural states. We have previously correlated the slower SRX rate to a folded back state, whereas the faster DRX rate appears to be correlated with a structure where the heads are not bound to the S2 tail (*Anderson et al., 2018*; *Rohde et al., 2018*).

Here, we performed the single-turnover ATPase experiment with mutant and WT 25-hep myosins. Surprisingly, the pliant region mutations showed no significant reduction in the SRX state (*Figure 7A–D and G*; WT = 57 ± 10% (mean ± SD); D778V = 63 ± 10%, p=0.20; L781*P* = 61 ± 11%, p=0.42; and

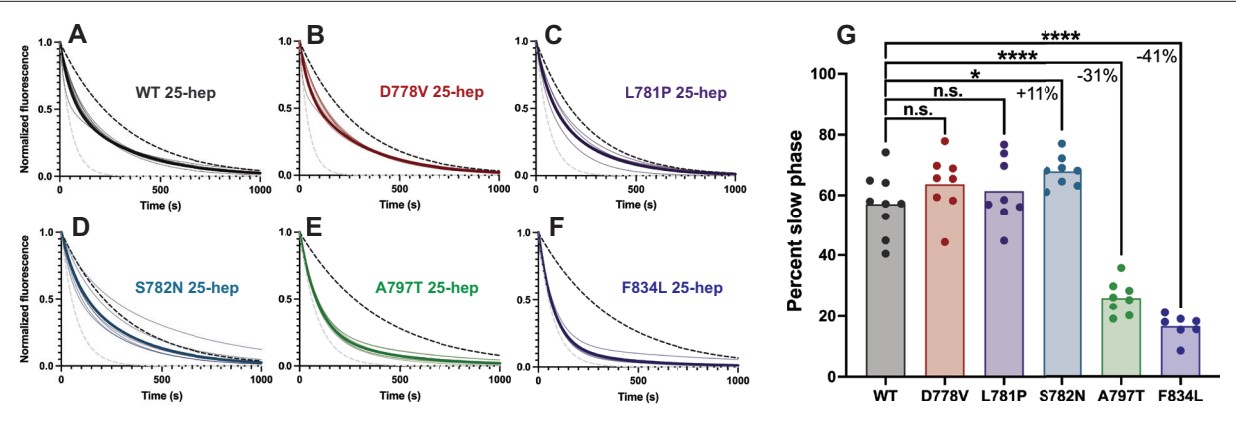

**Figure 7.** Single ATP turnover kinetics for 25-hep myosin lever arm mutants compared to wild type (WT). (**A–F**) Single mant-ATP turnover curves for WT 25-hep and each mutant. Thin curves show curves fitted to a biexponential decay and normalized to the fitted $Y_0$=1.0 and plateau=0.0 from each replicate, while thick lines show average fitted curves across all replicates. Note that in several cases, the average curve obscures individual replicate curves underneath. The dotted black line represents a simulated single-exponential decay with the slow rate of the average curve, and the dotted gray line represents a simulated single-exponential decay with the fast rate of the average curve. (**G**) Percent slow phase of multiple replicates of each mutant. Pairwise comparisons show that only A797T and F834L significantly reduce the super relaxed state (SRX) state, resulting in a 31 ± 4% and 41 ± 4% (mean ± SEM) reduction in slow phase, respectively. S782N resulted in an 11 ± 4% (mean ± SEM) increase in SRX. * indicates p≤0.05, **** indicates p≤0.0001.

The online version of this article includes the following source data and figure supplement(s) for figure 7:

**Source data 1.** Fast and slow rates and percent slow turnover for all 25-hep single turnover experiments.

**Figure supplement 1.** Representative single ATP turnover results for 25-hep myosins.

**Figure supplement 1—source data 1.** Representative single turnover data for a single technical replicate of each WT and mutant 25-hep.

**Figure supplement 2.** Single ATP turnover kinetics for wild type (WT) vs mutant 2-hep myosins.

**Figure supplement 2—source data 1.** Fast and slow rates and percent slow turnover for all 2-hep single turnover experiments.

S782N = 68 ± 5%, p=0.018; *Figure 7—figure supplement 1*). In fact, S782N showed a statistically significant, though functionally small, increase in the SRX phase. This unanticipated result is in opposition to the actin-activated ATPase result comparing the 2-hep vs 25-hep constructs, which suggested that the autoinhibition was disrupted. In contrast, the light chain-binding region mutations did show a statistically significant reduction in SRX compared to WT (*Figure 7E–F and G*; A797T = 26 ± 6%, a 31% decrease, p<0.0001; and F834L = 16 ± 4%, a 41% decrease, p<0.0001), in keeping with their increases in 25-hep actin-activated ATPase rates. None of the mutations affected the rates of either the fast or slow phases (*Supplementary file 2*).

As a control, we assayed the single-turnover ATPase rates of the 2-hep constructs, and we did not see any statistically significant changes in the percent SRX turnover for any of the mutations as compared to WT 2-hep (*Figure 7—figure supplement 2*; WT = 21 ± 5%; D778V = 16 ± 9%, p=0.19; L781*P* = 23 ± 2%, p=0.23; S782N = 21 ± 4%; p=0.99; A797T = 18 ± 6%, p=0.08; F834L = 26 ± 5%, p=0.12 [mean ± SD]). This suggests that the increased SRX turnover for both WT and the pliant mutations is S2 tail-dependent and not based on any changes specific to the motor domain. In contrast, the A797T and F834L 2-heps had a similar fraction of SRX turnover as compared to the 25-hep constructs, suggesting that these mutations virtually extinguish the S2 tail-dependent autoinhibition measured by the single-turnover assay.

## Discussion

The myosin lever arm performs one of the most important functions of the myosin chemomechanical cycle: it amplifies the transduction of the chemical energy of ATP hydrolysis into physical motion, allowing for myosin's motor function. This crucial role explains why the lever arm is both highly conserved and a hotspot for HCM-causing mutations. Here, we have characterized five such mutations spanning the pliant region and the light chain-binding regions, giving insights into the role of the lever

arm in generating force and power output, as well as its role in myosin S2 tail-based autoinhibition. The impacts of the mutations on myosin function segregate them naturally into two categories: the light chain-binding region mutations, A797T and F834L, behave very differently from the pliant region mutations, D778V, L781P, and S782N.

The functional changes in myosin's motor activity caused by the pliant region mutations were dramatic and ran the full spectrum of potential outcomes: D778V increased actin-activated ATPase of the 2-hep construct along with in vitro motility velocity and $k_0$, L781P decreased motility velocity, $k_0$, and step size, and S782N did not result in changes in any parameter except the force sensitivity of the detachment rate $\delta$. That these mutations can have such varied and substantial effects on myosin activity speaks to the importance of the pliant region in coupling the activity of the lever arm to the rotation of the converter domain.

One feature that was shared among the pliant region mutations was a reduction in the force sensitivity $\delta$. Reduced force sensitivity suggests a mechanism where mutations partially uncouple the force perceived at the ATP-binding site from the force applied at the anchor point at the C-terminus of the lever arm. We previously observed a decrease in $\delta$ for the P710R mutation, which is located in between the SH helices and the converter domain approximately 20 Å from the pliant region (**Vander Roest et al., 2021**). Pliant region mutations may likewise reduce rigidity, thus uncoupling the force transduction and reducing $\delta$. The term 'pliant region' itself comes from the varied positions of this region observed in different crystal structures (**Houdusse et al., 2000**; **Gourinath et al., 2003**), suggesting that its compliance or ability to assume multiple conformations is an important feature of its function; thus, any alteration to the rigidity of this region would likely have strong impacts on myosin function in sarcomeres. It is possible that reductions in $\delta$ as a result of pliant region mutations could impact contraction and/or relaxation kinetics in patients with these specific mutations.

Additionally, divergent changes resulting from the pliant region mutations may be a function of the identities of the mutations themselves. For example, the L781P mutation introduces a helix-breaking proline into a continuous α-helix. If the lever arm α-helix is destabilized, this could explain the observed decrease in step size for the L781P mutation: a more bent lever arm would generally produce a lower step size. Additionally, the D778V mutation replaces a highly polar amino acid with a more hydrophobic one. This mutation dramatically increases the detachment rate, suggesting that amino acids in the pliant region, which are very distant from the actin-binding domain, can play a role in the highly allosteric process of actin detachment. Conversely, the S782N mutation has comparatively few impacts on the kinetics of the myosin motor, suggesting that some mutations in the pliant region may be well tolerated in the context of motor domain function.

The sum total of the effects of pliant region mutations on motor activity (outside of potential impacts on autoinhibition) is reflected in the changes in calculated average power output (**Figure 4C**), which takes into account actin-activated ATPase rate $k_{cat}$, force-dependent detachment rate $k_{det}(F)$, force-independent detachment rate $k_0$, and step size $d$. For the pliant region mutations, D778V appears to be hypercontractile (particularly at high resistive forces), L781P appears to be hypocontractile, and S782N is within error of WT. A prominent model in the field suggests that HCM mutations increase myosin activity in ensemble, resulting in hypercontractility at the cellular level (**Spudich, 2019**). The magnitude of the increases in D778V's actin-activated ATPase activity and motility velocity was comparable to the early-onset mutations D239N and H251N (**Adhikari et al., 2016**) and not far from the changes observed for I457T, which had the largest changes we've observed to date (**Adhikari et al., 2019**). Thus, it is possible that these increases alone could lead to hypercontractility in a cellular context. However, the same cannot be said of L781P and S782N, where a different rationale would have to apply, given that these mutations resulted in either decreased or no change in contractility, respectively. Thus, we additionally considered the possibility that these mutations might impact myosin S2 tail-based autoinhibition.

The pliant region mutations showed a unique phenotype where they had a smaller reduction between the actin-activated ATPase rate of the 2-hep vs the 25-hep constructs as compared to WT, suggesting reduced autoinhibition, but did not show any statistically significant changes in the SRX/DRX ratio in the single turnover ATPase assay. We have used these two assays to study a number of mutations to date (**Adhikari et al., 2019**; **Sarkar et al., 2020**; **Vander Roest et al., 2021**), and all previously studied mutations had shown concordant behavior in these two assays. One way to interpret this data is that the actin-activated ATPase of 25-hep suggests that autoinhibition is disrupted by

the mutations, but only in the presence of actin, such that the single turnover ATPase is unchanged. Previous data supports a model where the presence of actin can influence and disrupt myosin autoinhibition (*Rohde et al., 2018*), so it stands to reason that mutations could specifically affect this activity. This model would suggest that in the in vivo context, myosin activity would be increased specifically during contraction, when actin is available for myosin binding, thus increasing force production, but during relaxation, when actin-binding is blocked by tropomyosin, there would be the same proportion of SRX compared to WT. It is clear from structures of smooth muscle myosin in the folded state that the conformation of the pliant region is different in the two heads (*Scarff et al., 2020*; *Yang et al., 2020*; *Heissler et al., 2021*): the blocked head pliant region takes on a bent conformation, while in the free head pliant region is much straighter. Thus, logically, mutations could destabilize the IHM by changing the available positions of the pliant region, leading to the reduced autoinhibition we observed and thus hypercontractility. Alternatively, it is possible that this study of the pliant region mutations may be limited by isolating the myosin outside of the context of sarcomeres, where structural constraints or additional proteins (such as myosin binding protein-C) could impact myosin activity and/or the formation of the autoinhibited state. Regardless, future experiments introducing pliant region mutations into a cardiomyocyte model would be useful for helping to test whether these mutations indeed result in net hypercontractility at the cellular level, and if so, how that would arise and lead to cellular hypertrophy.

In contrast to the pliant region mutations, the light chain-binding region mutations both had very little or no impact on myosin's in vitro motility velocity or actin-activated ATPase activity of the 2-hep construct. However, both mutations had clear impacts on myosin's S2 tail-based autoinhibition, reflected by both an increase in the S2 tail-containing 25-hep actin-activated ATPase rate and a decrease in the proportion of SRX. Thus, it is likely that the light chain-binding regions of the lever arm play an important role in myosin S2 tail-based autoinhibition. Most myosin mutations that have been investigated in the context of myosin autoinhibition have been in regions thought to be directly involved in forming structurally necessary contacts for the IHM state, including mutations in both the myosin 'mesa' region and the S2 tail (*Nag et al., 2017*; *Adhikari et al., 2019*), which likely interact with each other in the IHM, as well as mutations at the putative head-head interface (*Sarkar et al., 2020*). In contrast, A797T and F834L do not appear to be involved in such contacts in modeled structures of β-cardiac myosin in the IHM state (*Nag et al., 2017*; *Alamo et al., 2017*; *Robert-Paganin et al., 2018*). Thus, if the defects in autoinhibition that we observed for A797T and F834L indeed indicate reductions in the IHM structural state, these mutations are likely to affect the IHM structure by an allosteric mechanism. A simple model could be that the A797T and F834L mutations affect the position or rotation of the light chains about the lever arm. While our data shows that the light chains were still able to load onto the lever arm at the proper ratios, it is possible that specific light chain positioning is required to access the folded state. Indeed, in structures of smooth muscle myosin in the folded state (*Scarff et al., 2020*), it was shown that the RLC in particular took a position such that its phosphorylation sites were very close together in the IHM state, which the authors hypothesized created energetically unfavorable charge-charge interactions when the sites were phosphorylated (phosphorylation has been shown to move heads out of the autoinhibited state). Additionally, it was shown that a specific interaction of the C-termini of the RLCs with the hook joint (near F834L) on both the free and blocked heads may be important for formation of the IHM (*Heissler et al., 2021*). Thus, positioning of the light chains appears to be important for forming the folded state.

Alternatively, it is possible that the light chain-binding region mutations do not affect the positioning of light chains; instead, they could affect the structure of the lever arm itself in ways that prevent autoinhibition. It is apparent from the modeled structures of β-cardiac myosin in the IHM state that the lever arm may form an unusual conformation to allow the heads to fold back. In all three structures, the myosin heads and S2 tails overlay remarkably well; however, the lever arms vary dramatically (*Nag et al., 2017*; *Alamo et al., 2017*; *Robert-Paganin et al., 2018*; *Figure 8A-C*). Notably, in all three modeled structures, the lever arm's α-helix is broken; this differs from known structures of the folded state in other myosins, where the α-helix is largely unbroken (*Scarff et al., 2020*; *Yang et al., 2020*; *Heissler et al., 2021*; *Figure 8D–F*). This suggests that the homology modeled structures of human β-cardiac myosin may differ from the real structure (especially in the lever arm region) and emphasizes the need for an atomic-resolution structure of human β-cardiac myosin in the 'off' state. Thus, it is possible that a specific or unusual conformation of the lever arm might be required to allow

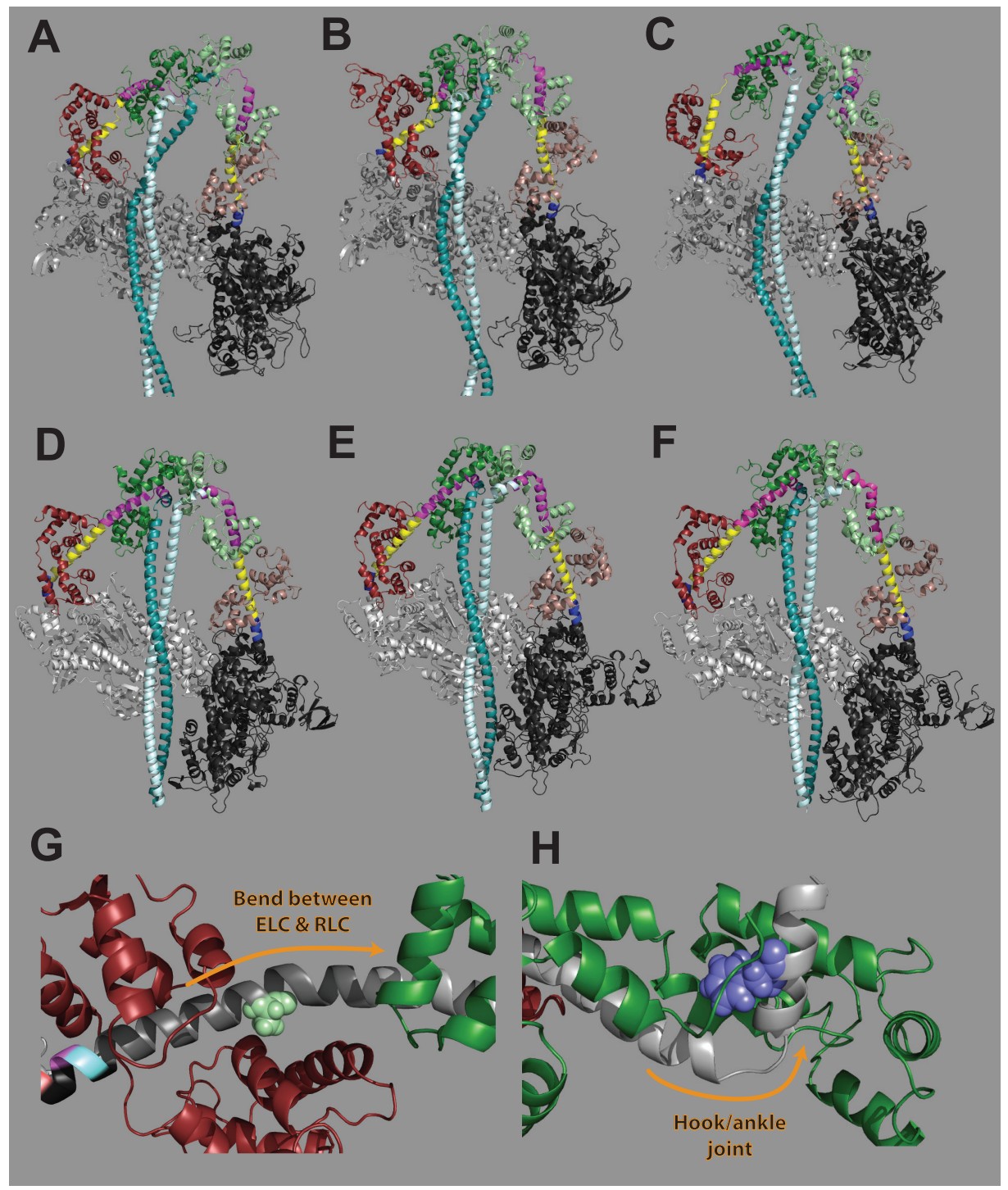

**Figure 8.** Contributions of lever arm position to the folded state structure. (**A–F**) Homology modeled structures of human β-cardiac myosin in the interacting heads motif (IHM) structural state from (**A**) (**Alamo et al., 2017**), (**B**) (**Nag et al., 2017**), and (**C**) (**Robert-Paganin et al., 2018**) and cryo-EM solved structures of smooth muscle myosin in the IHM structural state from (**D**) (**Scarff et al., 2020**), (**E**) (**Yang et al., 2020**), and (**F**) (**Heissler et al., 2021**). In each structure, the pliant region is colored yellow, the essential light chain (ELC)-binding region is colored blue, and the regulatory light chain (RLC)-binding region is colored pink. The myosin heads are in gray, ELCs are in dark red and brown, RLCs are in green, and subfragment 2 tail regions are in cyan. Homology modeled structures (**A–C**) show a markedly different lever arm structure as compared to experimentally determined structures (**D–F**). (**G**) A797 (light green) is located in the region where the lever arm bends between the ELC and RLC. This region has been previously implicated in formation of the IHM (**Houdusse and Cohen, 1996**). (**H**) F834 (blue) is located very near to the hook or ankle joint at the end of the lever arm. This joint has been previously shown to be important in the formation of the IHM (**Brown et al., 2011**). (**G**) and (**H**) are from a homology modeled structure of human β-cardiac myosin in the prestroke state (**Farman et al., 2014**; see **Figure 1A**).

the IHM structure, and mutations might preclude specifically those conformations. For example, in molluscan myosin, where the folded back state has been extensively studied, the lever arm plays a key role in the formation of the folded back state, and a few regions in particular have been identified as important. The first is the slightly bent region of the lever arm directly in between the two light chains; in molluscan myosin, this is where $Ca^{2+}$ directly binds to myosin and activates it from the folded back autoinhibited state (*Houdusse and Cohen, 1996*). Coincidentally, the A797T mutation is very near to that slight bend (Fig. S7G). Another region that has been identified as important in molluscan myosin is the 'hook' or 'ankle' joint near the C-terminus of the lever arm (*Pylypenko and Houdusse, 2011*; *Brown et al., 2011*). This feature is common across the myosin II class and can bend to very different angles, which is thought to help allow for the folded back conformation. The F834L mutation is very close to that ankle joint, and thus could affect the conformation about that joint (Fig. S7H). In any case, a high-resolution structure of human β-cardiac myosin in the IHM state will be useful to further understand the structural requirements of the lever arm for autoinhibition.

In summary, this study allowed for a detailed analysis of lever arm function, specifically focusing on how HCM-causing mutations in both the pliant and light chain-binding regions of the lever arm alter myosin activity. While the light chain-binding region mutants clearly showed a phenotype of reducing myosin autoinhibition, potentially by disrupting the folded back IHM conformation, the pliant region mutants had a more complicated phenotype that begs further investigation. With a high density of mutations across a very small region of myosin, the pliant region seems to play an intriguing role in both transducing force from the motor domain to the lever arm and potentially forming the IHM structure. Future research into the lever arm region, particularly using in vivo models, could further clarify the role of the lever arm in both force transduction and formation of the folded state.

## Materials and methods

**Key resources table**

| Reagent type (species) or resource | Designation | Source or reference | Identifiers | Additional information |
|---|---|---|---|---|
| Gene (*Homo sapiens*) | MYH7 | NCBI Gene | Gene ID: 4625 | |
| Gene (*H. sapiens*) | MYL3 | NCBI Gene | Gene ID: 4634 | N-terminal FLAG-TEV tag added |
| Gene (*H. sapiens*) | MYL2 | NCBI Gene | Gene ID: 4633 | N-terminal His-TEV tag added |
| Strain, strain background (*Escherichia coli*) | BJ5183-AD-1 | Agilent | 200,157 | |
| Strain, strain background (*E. coli*) | Rosetta (DE3) pLysS | Sigma-Aldrich | 70,956 | |
| Cell line (*H. sapiens*) | HEK 293T | ATCC | CRL-3216 | |
| Cell line (*H. sapiens*) | C2C12 | ATCC | CRL-1772 | |
| Transfected construct (*H. sapiens*) | pAdEasy-1-MYH7 sS1 | 10.1073/pnas.1309493110 | | |
| Transfected construct (*H. sapiens*) | pAdEasy-1-MYH7 2hep | 10.1038/nsmb.3408 | | |
| Transfected construct (*H. sapiens*) | pAdEasy-1-MYH7 25hep | 10.1038/nsmb.3408 | | |
| Biological sample (Bos taurus) | Bovine cardiac acetone powder | Pelfreez | 57,195 | |
| Recombinant DNA reagent | pShuttle-CMV- MYH7 sS1 | 10.1073/pnas.1309493110 | | |
| Recombinant DNA reagent | pShuttle-CMV- MYH7 2hep | 10.1038/nsmb.3408 | | |
| Recombinant DNA reagent | pShuttle-CMV- MYH7 25hep | 10.1038/nsmb.3408 | | |
| Sequence-based reagent | D778V S | This paper | Mutagenesis primer | GGAGGAAATGAGGGTCGAGAGGCTGAGCC |
| Sequence-based reagent | D778V AS | This paper | Mutagenesis primer | GGCTCAGCCTCTCGACCCTCATTTCCTCC |

*Continued on next page*

*Continued*

| Reagent type (species) or resource | Designation | Source or reference | Identifiers | Additional information |
|---|---|---|---|---|
| Sequence-based reagent | L781P S | This paper | Mutagenesis primer | GATGATGCGGCTCGGCCTCTCGTCCCT |
| Sequence-based reagent | L781P AS | This paper | Mutagenesis primer | AAGGACGAGAGGCCGAGCCGCATCATC |
| Sequence-based reagent | S782N S | This paper | Mutagenesis primer | TGAGGGACGAGAGGCTGAACCGCATCATC |
| Sequence-based reagent | S782N AS | This paper | Mutagenesis primer | GATGATGCGGTTCAGCCTCTCGTCCCTCA |
| Sequence-based reagent | A797T S | This paper | Mutagenesis primer | GTACTCCATTCTGGTGAGCACACCTCGGG |
| Sequence-based reagent | A797T AS | This paper | Mutagenesis primer | CCCGAGGTGTGCTCACCAGAATGGAGTAC |
| Sequence-based reagent | F834L S | This paper | Mutagenesis primer | CTGGATGAAGCTCTACTTAAAGATCAAGCCGCTG |
| Sequence-based reagent | F834L AS | This paper | Mutagenesis primer | CAGCGGCTTGATCTTTAAGTAGAGCTTCATCCAG |
| Software, algorithm | FAST | 10.1016 /j.celrep.2015.04.006 | | Filament tracking and velocity measurement software |

## Protein expression and purification

Recombinant human β-cardiac myosin constructs described within, including sS1, 2-hep (short-tailed), and 25-hep (long-tailed), were purified as described previously (**Nag et al., 2017**; **Sommese et al., 2013**) with some minor modifications. Heavy chain myosin (*MYH7*) was co-expressed with human ELC (*MYL3*) containing an N-terminal FLAG tag followed by a TEV protease site in the differentiated mouse myoblast C2C12 cell line (ATCC) using adenovirus generated in HEK293T cells (ATCC) using the AdEasy Vector System (Qbiogene Inc, Carlsbad, CA, USA). The sS1 construct used in this study contained a C-terminal eGFP tag, while the 2-hep and 25-hep constructs contained both eGFP and PDZ C-peptide on their C termini, respectively. C2C12 cells were infected with adenovirus constructs 48 hr after differentiation and harvested 4 days after infection in a lysis buffer containing 50 mM NaCl, 20 mM $MgCl_2$, 20 mM imidazole pH 7.5, 1 mM EDTA, 1 mM EGTA, 1 mM DTT, 3 mM ATP, 1 mM PMSF, 5% sucrose, and Roche protease inhibitors. Cells were then immediately flash frozen in liquid nitrogen. Note that this lysis buffer composition differs from our previously published methods—salt concentration and sucrose are lowered and $MgCl_2$ concentration is raised to encourage the native mouse myosin to form filaments, reducing contamination. Pellets were stored up to 6 months at –80°C, then thawed at room temperature for purification. HEK293 and C2C12 cell lines were tested for mycoplasma contamination using the Mycoalert plus kit (Lonza).

Cells were lysed with 50 strokes of a dounce homogenizer and clarified by spinning at 30,000 RPM in a Ti-60 fixed angle ultracentrifuge rotor for 30 min. Supernatant was bound to anti-FLAG resin for 1–2 hr at 4°C. The resin was then washed with a wash buffer containing 150 mM NaCl, 5 mM $MgCl_2$, 20 mM imidazole pH 7.5, 1 mM EDTA, 1 mM EGTA, 1 mM DTT, 3 mM ATP, 1 mM PMSF, 10% sucrose, and Roche protease inhibitors. For both the 2-hep and 25-hep constructs, native mouse RLC was depleted with a buffer containing 20 mM Tris pH 7.5, 200 mM KCl, 5 mM CDTA pH 8.0, and 0.5% Triton-X-100 for 1 hr at 4°C, and human RLC (purified from *Escherichia coli* as previously described [**Nag et al., 2017**]) was added to the resin with wash buffer and incubated for >2.5 hr at 4°C. The resin was then incubated overnight with TEV protease at 4°C to cleave the ELC-myosin complex off of the resin. The next day, the supernatant was further purified using a HiTrap Q HP column on an fast protein liquid chromatography (FPLC) with a gradient of 0–600 mM NaCl over 20 column volumes in a buffer containing 10 mM imidazole pH 7.5, 4 mM $MgCl_2$, 10% sucrose, 1 mM DTT, and 2 mM ATP. Pure fractions (determined by Coomassie staining on 10% SDS PAGE) were collected and concentrated to 5–50 uM using Amicon Ultra 0.5 mL centrifugal filters with a 50 or 100 kDa cutoff, which aids in removing any unbound ELC or RLC. The myosin was then used directly for ATPase assays, buffer exchanged for single turnover assays (described below), or flash frozen in liquid nitrogen for in vitro motility or optical trapping experiments.

## Deadheading

For further analysis using in vitro motility or optical trapping, the myosin was subjected to a 'dead-heading' procedure to remove any myosin that bound irreversibly to actin. After thawing on ice, myosin was first incubated with a >10× excess of F-actin for 5 min on ice. 2 mM ATP was then added

to the mixture, and it was further incubated for 3 min on ice. The F-actin was then pelleted by ultra-centrifugation at 95 K RPM in a TLA-100 rotor, and the supernatant containing active myosin was collected and used. This procedure was applied to all 2-hep myosin samples used for in vitro motility, while we found empirically that it was only necessary to deadhead the D778V sS1 myosin for optical trapping (the WT and other mutants were not deadheaded). This is because in a single molecule experiment, a 'dead' myosin head does not generate useful data; instead, it binds to the actin irreversibly and destroys the actin dumbbell. This rarely occurred for the WT, L781P, and S782N myosins, but was more frequent for D778V. Thus, we deadheaded only D778V to reduce dumbbell loss.

## In vitro motility

Motility measurements of 2-hep WT and mutant myosins were collected as described previously (*Aksel et al., 2015*). Multichannel flow chambers were constructed on microscopy slides using double-sided tape and coverslips coated in 0.1% nitrocellulose/0.1% collodion in amyl acetate. SNAP-PDZ (purified from *E. coli* as described previously [*Aksel et al., 2015*]) was first flowed into each channel at a concentration of 3 µM in assay buffer (AB: 25 mM imidazole pH 7.5, 25 mM KCl, 4 mM MgCl₂, 1 mM EGTA, and 10 mM DTT) and incubated for 2 min at room temperature. Then, each channel was blocked with assay buffer plus 1 mg/mL BSA (ABBSA) twice for 2 min each. Myosin, diluted to 50–100 nM in ABBSA, was then flowed into each channel and incubated for 2 min. Each channel was then washed with ABBSA, then loaded with final GO buffer containing 2 mM ATP, 2.5 nM TMR-phalloidin-labeled actin, and an oxygen-scavenging system (0.4% glucose, 0.11 mg/mL glucose oxidase, and 0.018 mg/mL catalase) in ABBSA.

The slide was then imaged on a Nikon Ti-E inverted microscope with a 100 × TIRF objective at a rate of 2 Hz with an exposure of 300 ms for 30 s on an Andor iXon + EMCCD camera. Each channel was imaged in three separate frames, and that data was combined for analysis. Movies were analyzed using Fast Automated Spud Trekker (*Aksel et al., 2015*) for filament tracking and velocity measurement using the following parameters: window size n=5, path length *P*=10, percent tolerance pt=20, and minimum velocity for stuck classification minv=80 nm/s. Filtered mean velocity was used as unloaded velocity. At least five replicates of each mutant were performed across 13, 3, 3, 4, 3, and 4 independent protein preparations for WT, D778V, L781P, S782N, A797T, and F834L 2-heps, respectively (biological replicates), with additional technical replicates for mutants with smaller or no difference from WT (to confirm little or no variation from WT). Temperature varied from 21 to 23°C during imaging; these temperature variations result in some variation in velocity between slides, but every mutant channel was imaged on the same slide at the same temperature as a WT control. Each mutant was then normalized to the WT control from the same slide, which negates the effect of slight

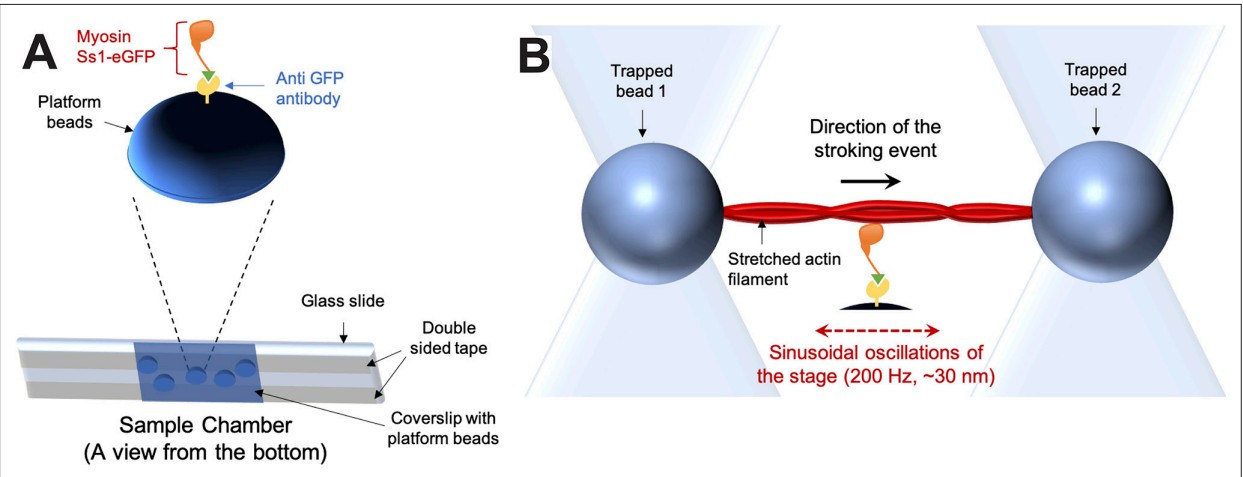

**Figure 9.** Technical details of the dual-beam optical trap experiment. (**A**) The sample chamber (bottom) is shown in an orientation suitable for an inverted microscope. On top, the typical arrangement of the protein complexes on top of a platform bead is depicted. (**B**) A typical stroking event—as expected during actin-myosin interaction in a harmonic force spectroscopic (HFS) setup—is depicted. In a standard HFS experiment, the stage oscillates, resulting in a variety of assistive or resistive external load forces applied to the stroking myosin.

variations in temperature. p-Values were determined using a paired t-test comparing mutant velocities to their paired WT control velocities.

## Optical trapping

To obtain load-dependent detachment rates of single myosin-actin interaction events, HFS measurements were performed in a dual-beam optical trap. The details of the instrumental setup and the method are similar to what has been previously described (*Liu et al., 2018*; *Sung et al., 2015*; *Sung et al., 2017*). The sample chambers for this experiment are made by sticking a No. 1.5 coverslip onto a 1 mm thick microscope slide with the use of double-sided tape (*Figure 9A*, bottom). These coverslips are spin-coated with silica beads (diameter ~1.6 micrometer, Bangs Laboratories, Fishers, IN, USA) and then with an amyl acetate solution containing 0.1% nitrocellulose and 0.1% collodion. After forming the chamber, the surface of the coverslip and the platform beads are functionalized by flowing anti-GFP antibody (product #ab1218, Abcam, Cambridge, United Kingdom) solution in AB buffer (25 mM imidazole at pH 7.5, 25 mM KCl, 4 mM MgCl$_2$, 1 mM EGTA, and 10 mM DTT) into the chamber for 2–3 min. The concentration of the anti-GFP antibody solution is kept low (1 nM) to ensure only a stochastic presence of the molecules on the surface. BSA (1 mg/mL) solution in AB buffer (ABBSA) is then flown into the chamber for 2–3 min to passivate/block the remaining uncovered glass surfaces. A 50 nM solution of GFP functionalized (C-terminal) short construct of either WT or mutants of human β-cardiac myosin (sS1-eGFP) in ABBSA buffer is then flown into the chamber for 2–3 min. These sS1-eGFP molecules are thus translationally immobilized on the surface by binding to the anti-GFP antibody (Fig. S7A, top). The unattached free myosin molecules are then washed away with ABBSA buffer. Finally, a solution containing filaments of 0.3 nM TMR-phalloidin-labeled biotinylated actin (Cytoskeleton, Denver, CO, USA), 0.4% glucose, 0.11 mg/mL glucose oxidase, 0.018 mg/mL catalase, 2 mM ATP, and 1-micrometer-diameter streptavidin-coated polystyrene beads (Bangs Laboratories) diluted (~5000 times) in ABBSA is flown in, prior to the sealing of the chamber with vacuum grease. This entire sample chamber preparation is done at 23°C.

The stiffnesses of the two optical traps were kept between 0.08 and 0.10 pN/nm for all experiments. Each trap was calibrated by trapping single polystyrene beads in them and then by using the power spectrum method, as described elsewhere (*Liu et al., 2018*; *Sung et al., 2015*; *Sung et al., 2017*). Corrections were made to rectify the effect of the anti-aliasing filter in the system. The contributions from the surface effects were also corrected during calibration. At this point in the experiment—after calibration—each trap contained one streptavidin-coated polystyrene bead. Next, an actin filament was snared between the two trapped beads using a biotin-streptavidin linkage by moving the chamber (i.e. the microscope stage) with respect to the position of the optical traps in 3D. The filament is then stretched by pulling the two beads from their ends (by steering the optical traps away from each other) to form a 'dumbbell' (*Figure 9B*). At this point in the experiment, the actin filament is stretched and held in the solution. On the surface, the myosin molecules (in the presence of ATP) are ready to initiate stroking events upon actin interaction. The actin dumbbell is then lowered toward the platform beads on the surface (keeping the stage under 200 Hz oscillation) anticipating potential interactions with single myosin molecules (*Figure 9B*). The stage oscillation imparts different amounts of assistive or resistive external forces during each stroking event based on the stochastic binding of myosin. In the time-trace data, the myosin-actin interactions and stroking events are identified by an expected change in the phase and the amplitude of the bead-oscillation in both the traps (*Liu et al., 2018*). Upon detachment after the stroking events, the oscillation in the position of the two trapped beads returns to its initial value. Several stroking events for each myosin molecule are recorded. These stroking events are then identified and binned based on the extent of external force. Then, using maximum likelihood estimation, detachment rates for every force range are obtained from the durations of the events for each molecule (*Liu et al., 2018*). The external load/force (F) dependent change in the detachment rates ($k_d$) is exponential in nature:

$$k_d\left(F, \Delta F\right) = k_0 I_0 \left(\frac{\Delta F \delta}{k_B T}\right) e^{\frac{-F\delta}{k_B T}} \tag{7}$$

where $k_B$ is the Boltzmann constant, T is the temperature, $k_0$ is the detachment rate at zero external force, $I_0$ is a correction for the harmonic force with $\Delta F$ amplitude, and $\delta$ is the measure of force sensitivity of the myosin molecule. $k_0$ and $\delta$ are the parameters that vary in mutant

myosins as compared to WT. These values can be obtained by fitting the spread of the $k_d$ values at different external forces with the *equation S1*. During each stroking event, the positions of the dumbbells are shifted accordingly and therefore, the step sizes for individual myosin molecules can also be obtained by analyzing the same HFS data (*Vander Roest et al., 2021*). In the HFS experiments, however, the presence of several compliant-elements—e.g., rotation of the beads, biotin-streptavidin linkage between the beads and the actin, surface attachment of the myosin etc.—makes the step size measurement complicated. Our reported data of step-sizes do not have compliance correction and the values of the step-sizes are, therefore, smaller than what is expected for myosin's working stroke. The data reported in this paper are gathered from the inter-actions of multiple different individual myosin molecules with different actin dumbbells (technical replicates) during independent experiments done over several different days with two separate protein preparations for each WT and mutant protein (biological replicates). p-Values for WT vs each mutant $k_0$, $\delta$, and step size were determined using an unpaired t-test with Welch's correction (correcting due to varying SDs by mutant), where each molecule was treated as an individual repli-cate. Molecules came from 2, 2, 2, and 3 unique protein preparations for WT, D778V, L781P, and S782N sS1, respectively (Fig. S4).

## Actin-activated ATPase assay

Actin-activated ATPase rates were measured using an NADH-coupled assay as previously described (*Vander Roest et al., 2021*; *De La Cruz and Ostap, 2009*). Actin was prepared as described previ-ously (*Spudich and Watt, 1971*) and dialyzed 4× into assay buffer: 5 mM KCl, 10 mM imidazole pH 7.5, 3 mM MgCl$_2$, and 1 mM DTT. Actin was then mixed with a 1:50–200 molar ratio of gelsolin (prepared from *E. coli* as described previously [*Dawson et al., 2003*]), mixed thoroughly, and incu-bated on ice for >30 min. In a clear 96-well plate with 100 uL final volume, actin × gelsolin was mixed with assay buffer to achieve 0–80 µM final concentrations and enough myosin to achieve 25 nM final concentration. To measure the basal ATPase rate, myosin was added without actin at final concen-trations of 75–125 nM. The plate was then incubated at room temperature for 10 min with constant shaking. To initiate the reaction, 20 uL of a 5× coupling solution containing 100 U/mL lactate dehy-drogenase (product #L1254, Sigma-Aldrich, St. Louis, MO, USA), 500 U/mL pyruvate kinase (product #500–20, Lee Biosolutions, Maryland Heights, MO, USA), 2.5 mM phospho(enol) pyruvate (Sigma P0564), 10 mM ATP, and 2 mM NADH (Sigma N8129). The plate was again incubated for 2–5 min at room temperature with shaking before reading absorbance at 340 nm every 15–30 s for 15–25 min. A standard curve of ADP from 0 to 300 µM was created to convert the absorbance values to concen-tration of ADP produced.

Rates for each concentration of actin were calculated from the slope of a plot of (ADP) produced over time. For each concentration of actin, technical triplicates were performed on the same plate using the same proteins. These rates were divided by the concentration of myosin in each well, plotted against the concentration of actin, and fitted to Michaelis-Menten kinetics to obtain the values in columns 4 and 5 of *Supplementary file 1*, where the error reported is the SE of the fit. Each mutant 2-hep and 25-hep was prepared in tandem with a paired WT 2-hep control, and technical triplicates were measured for two independent biological replicates, where biological replicates use proteins made from different C2C12 cell batches and prepared freshly on separate days. The two biological replicates are used to validate each other; we have previously observed that additional biological replicates do not typically differ when the two biological replicates match. Within technical tripli-cates, individual measurements were rejected if they differed from the other two values by >50%; this resulted in rejecting less than 4% of total measurements. This is necessary because actin is inherently viscous, sometimes resulting in pipetting errors that lead to erroneous rates.

To obtain plots in *Figures 2 and 5*, rates from one biological replicate were normalized to the $k_{cat}$'s of the same-day WT 2-hep and mutant 2-hep, respectively. A t-test was used to compare the WT 2-hep to the mutant 2-hep rates. To obtain the data in *Supplementary file 1*, each individual biolog-ical replicate was fitted to standard Michaelis-Menten kinetics to obtain $k_{cat}$ and $K_{app}$, where errors reported represent the SE of the fit. Errors reported in the four rightmost columns of *Supplemen-tary file 1* (mutant 2-hep/WT 2-hep $k_{cat}$ ratio, mutant 25-hep/mutant 2-hep $k_{cat}$ ratio, average mutant 2-hep/WT 2-hep $k_{cat}$ ratio, and average mutant 25-hep/mutant 2-hep $k_{cat}$ ratio) are calculated using SE propagation methods for ratios and averages, respectively, propagating the SE from the fit of $k_{cat}$.

## Single ATP turnover assays

To determine data reported in *Figure 6*, Fig. S2, and *Supplementary file 2*a single mant-ATP turnover assay was used as described previously (*Anderson et al., 2018*). Briefly, myosin was buffer exchanged 5× into assay buffer containing 100 mM KOAc, 10 mM Tris pH 7.5, 1 mM DTT, 4 mM $MgCl_2$, and 1 mM EDTA in a 50 or 100 kDa cutoff 0.5 µL Amicon filter. Myosin was then mixed with appropriate volumes of assay buffer containing 0 mM KOAc and 100 mM KOAc to achieve final salt concentrations of 5 mM (for 25-hep myosin) or 25 mM (for 2-hep myosin—2-hep does not show a dependence on salt concentration in this assay [*Anderson et al., 2018*]) KOAc and myosin concentrations of 200–900 nM with a 100 µL final assay volume. This was added to a 96-well black plate, where only a single well was measured at a time. 2'-(or-3')-O-(N-methylanthraniloyl) adenosine 5'-triphosphate (mant-ATP, Thermo-Fisher Scientific, Waltham, MA, USA) was serially diluted to concentrations of 5–15 µM in assay buffer and added to the myosin at a final concentration of 1×–1.2×. Within 10–20 s, excess unlabeled ATP (4 mM final concentration) was added to the myosin + mant-ATP mixture, and the fluorescent signal (470 nm Em/405 nm Ex) was measured every ~2 s for 16 min. The 'dead time' between adding unlabeled ATP and the first fluorescence measurement was recorded for each well and added to the time measured by the plate reader. Fluorescence signal vs time was plotted for each replicate and fitted to a five-parameter biexponential decay. Ambiguous fits were discarded. Average fast rates, slow rates, percent fast phase decay, and percent slow phase decay (with their SEMs) are presented in *Supplementary file 2*. For *Figure 6A–F*, representative curves for each protein were normalized to the fitted $Y_0 = 1.0$ and plateau value=0.0 and fitted again to a five-parameter biexponential decay. They are plotted alongside simulated single exponential curves that have the fast rates and slow rates, respectively. A range of 4–10 technical replicates of each protein construct were performed across 8, 3, 4, 3, 3, and 3 independent protein preparations for WT, D778V, L781P, S782N, A797T, and F834L 2-heps, respectively, and 4, 4, 5, 4, 5, and 4 independent protein preparations for WT, D778V, L781P, S782N, A797T, and F834L 25-heps, respectively (biological replicates). p-Values for WT vs each mutant were determined using unpaired t-tests across all technical and biological replicates.

## Light chain loading gel assay

Loading of the ELC and RLC was determined using denaturing SDS-PAGE. Before analysis, WT and mutant 2-hep myosins were buffer exchanged 5× in a 50 or 100 kDa 0.5 µL Amicon filter to remove any light chains that were unbound to the heavy chain. Myosin samples were loaded in a dilution series across the gel at 10 pmol, 5 pmol, 3 pmol, 2 pmol, and 1 pmol per lane. After separation by gel electrophoresis, the gel was stained in Coomassie, destained, and its fluorescence was scanned at 700 nm with a LI-COR Odyssey imaging system. Each band was quantified using Fiji (*Schindelin et al., 2012*). A plot of raw integrated density vs pmol protein loaded was generated for each light chain and the heavy chain of each protein sample. Non-linear points were removed (due to the much higher molecular weight of the heavy chain as compared to the light chains, the linear range does not fully overlap—e.g., the 10 pmol load is generally non-linear with the 5, 3, 2, and 1 pmol loads for the heavy chain), and a linear fit was generated for each light chain and the heavy chain of each protein sample separately. The slope of each light chain fit was divided by the slope of the heavy chain fit for each protein sample, respectively, to give raw data as presented for WT 2-hep in Fig. S1B. Expected ratios based on the molecular weights of the light chains and heavy chain are noted in the legend of Fig. S1, but Coomassie staining is biased by amino acid identity in each protein sample, thus it is not unusual that measured ratios deviate somewhat from expected values based on molecular weight alone. Fig. S1 shows light chain ratios for each mutant 2-hep normalized to their same-day WT 2-hep controls. Three biologically independent protein preparations were analyzed for each mutant along with nine biologically independent protein preparations for WT 2-hep. p-Values were determined using a paired t-test, where each mutant 2-hep was paired with its same-day WT 2-hep control.

## Acknowledgements

We wish to acknowledge Chao Liu for training, support, and commentary, especially with regards to the optical trapping experiments. We also thank Masataka Kawana, Neha Nandwani, Darshan Trivedi, and Alison Schroer Vander Roest in the Spudich group for useful comments. We also want to thank our collaborators Leslie Leinwand and Thomas Perkins and their research group members, who provided comments and support throughout the project. Funding: This work was funded by

NIH grants HL117138 and 2GM033289 to JAS and KMR. MMM was supported by Stanford Cellular and Molecular Biology Training Grant T32GM007276. DP was supported by an AHA postdoctoral fellowship.

## Additional information

### Competing interests

James Spudich: JAS is cofounder and on the Scientific Advisory Board of Cytokinetics, Inc, a company developing small molecule therapeutics for treatment of hypertrophic cardiomyopathy. The other authors declare that no competing interests exist.

### Funding

| Funder | Grant reference number | Author |
| --- | --- | --- |
| National Heart, Lung, and Blood Institute | Division of Intramural Research 1 R01 HL1171138-01A1/1550766 | James Spudich |
| National Institute of General Medical Sciences | 5 R01 GM033289-33A1 | James Spudich |
| National Institutes of Health | HL117138 | James Spudich Kathleen M Ruppel |
| National Institutes of Health | 2GM033289 | James Spudich Kathleen M Ruppel |
| Standford Cellular and Molecular Biology Training | T32GM007276 | Makenna M Morck |
| American Heart Association | Postdoctoral Fellowship | Divya Pathak |

The funders had no role in study design, data collection and interpretation, or the decision to submit the work for publication.

### Author contributions

Makenna M Morck, Conceptualization, Data curation, Formal analysis, Investigation, Methodology, Validation, Writing - original draft; Debanjan Bhowmik, Data curation, Formal analysis, Investigation, Methodology, Writing – review and editing; Divya Pathak, Investigation, Writing – review and editing; Aminah Dawood, Resources; James Spudich, Conceptualization, Funding acquisition, Writing – review and editing; Kathleen M Ruppel, Conceptualization, Formal analysis, Investigation, Project administration, Resources, Supervision, Writing – review and editing

### Author ORCIDs

Makenna M Morck ![ORCID] http://orcid.org/0000-0001-7561-892X
Kathleen M Ruppel ![ORCID] http://orcid.org/0000-0002-0971-5331

### Decision letter and Author response

Decision letter https://doi.org/10.7554/eLife.76805.sa1
Author response https://doi.org/10.7554/eLife.76805.sa2

## Additional files

### Supplementary files

• Supplementary file 1. Full actin-activated ATPase results. For each mutation, two independent biological experiments were performed with freshly prepared myosins, where each biological replicate was measured with technical triplicates (6 total replicates for each protein). For WT 2- vs 25-hep, five biological replicates were measured with technical triplicates (15 total replicates for each protein). The 2-hep and 25-hep mutant myosins were prepared in tandem with a WT 2-hep control for comparison, given that actin-activated ATPase results show a slight drift from day-to-day.

Results in the fourth and fifth columns, respectively, show the fitted values for $k_{cat}$ ($s^{-1}$) and $K_{app}$ (µM) ± SE of the fit for each biological replicate. The $k_{cat}$ ratio of mutant 2-hep/WT 2-hep and mutant 25-hep/mutant 2-hep for each independent biological replicate is shown in the sixth and seventh columns, respectively, where the error is propagated from SE of the fit for each measurement. In the rightmost column (average mutant 25-hep/mutant 2-hep kcat ratio), statistically significant differences for mutant ratios vs WT ratio are shown, where * indicates p≤0.05, ** indicates p≤0.01, and *** indicates p≤0.001.

• Supplementary file 2. Full single ATP turnover results. Single ATP turnover experiments were performed for both 2-hep and 25-hep myosins. Results show n, the number of independent experiments for each protein, average percent fast and slow phase ± SD, and average fast and slow rates ± SD. Stars denote statistically significant changes as compared to the WT control of the same protein construct, where * indicates p≤0.05 and **** indicates p≤0.0001.

• Transparent reporting form

## Data availability

All data generated or analysed during this study are included in the manuscript and supporting file; source data files have been provided for all data figures.

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
