## [Editor Report]

This paper explores several mutations lying in the lever arm region of cardiac myosin. Using a diverse array of biochemical, kinetic and biophysical techniques the authors show that the individual mutations can have many effects, not only on the kinetic properties of the ATPase cycle of the myosin, but also on the ability of the myosin to adopt an auto-inhibited conformation involving regions of the myosin outside of the motor domain. It shows the value of examining disease causing mutations of myosin using a variety of techniques.

---

## [Decision Letter]

**Decision letter after peer review:**

Thank you for submitting your article "HCM mutations in the pliant and light chain-binding regions of the lever arm of human β-cardiac myosin have divergent effects on myosin function" for consideration by *eLife*. Your article has been reviewed by 4 peer reviewers, including James R Sellers as Reviewing Editor and Reviewer #2, and the evaluation has been overseen by Anna Akhmanova as the Senior Editor. The following individual involved in review of your submission has agreed to reveal their identity: E Michael Ostap (Reviewer #3).

Essential revisions:

Thank you for submitting your manuscript to *eLife*. It was examined by three expert reviews and myself. The general conclusion is that the work is thorough, significant and, for the most part, very well documented. All of the reviewers felt that if their concerns can be adequately addressed, the work merits publication.

There were three major concerns. The first two deals with whether you have appropriate statistical power to back up your claims. One reviewer was concerned that the number of biological replicates is only two and this is magnified by the second concern which is the significant variability in the values reported in Table S1 for the WT sample. For example, the kcat varies from 4.64 to 6.04 /S in the two WT biological replicates and their Km values show even more variability for that replicate. Since all the data are normalized to the WT control, this raises concerns. In the manuscript you attribute this to "assay variation", but this degree of difference is surprising and reinforces the need for a third replicate. All of the reviewers appreciate the work involved in prepping and assaying six proteins and so we did not ask this lightly. Can you distinguish between assay variation and the case where the first WT replicate simply had less active motors? The data reported for the mutants are much tighter, but even their variation makes us wonder whether the small differences reported as deviations from WT are valid. Please either provide data from another replicate or give us strong reasons why we should not be concerned over this.

The third major concern deals with the optical trapping results. One reviewer commented that you should show some representative primary data showing single interactions and that a plot of displacement distributions would be important to see, given that these mutations are in the lever arm. More detail on the assumptions that go into the fittings of the harmonic force-spectroscopy data is also asked for.

In addition, each reviewer had specific items that should be addressed.

*Reviewer #2 (Recommendations for the authors):*

P.22 Figure S3 Legend. Authors in discussing A797 stated "This region has previously been implicated in the formation of the folded state" and later on that F834 which is in the ankle joint that "this joint has been previously shown to be important in autoinhibition". I'm assuming that the folded state and autoinhibition are both referring to the IHM state. Also, no reference is given for these statements.

Table S1 and P.7, line 5. The actin titrations of the myosin's ATPase activity aretypically fit to a rectangular hyperbola such as that used for analysis of the substrate dependence of enzyme kinetics with the Michaelis-Menten formalism. However, actin is not a substrate for myosin and so the term describing the actin concentration at half maximal Vmax should not be called a Km. Many authors use Kactin or Katpase to describe this value.

*Reviewer #3 (Recommendations for the authors):*

The authors need to better outline the assumptions that go into the fitting of the harmonic force-spectroscopy data. Would the analysis find non-linear mechanical behaviors of the lever caused by the mutation? What if the force-dependence didn't follow Bell's-Law like kinetics? I think the technique is totally appropriate for the study, but its limitations should be outlined.

What are the uncertainties on the plots in Figure 3A and Figure 4.

The Plots in Figures 2 and 5 are plotted as normalized values. Why not plot the ATPase values? It would be concerning if this is because there is too much variability from the assays.

It would be useful for Table S1 to report the mutant25-*Hep*/WT25-*Hep* kcat ratio.

*Reviewer #4 (Recommendations for the authors):*

The critique below list specific questions for the authors to consider, ordered as they occur in the manuscript. Quotes indicate text in the manuscript on the indicated line.

Page 2 line 8. Regarding "Frequent cause of hypertrophic…" can the authors state how frequent is frequent?

Page 2 line 12. Please avoid ambiguous and qualitative terms such as "relatively high" and "frequent" (see previous comment) as they are… ambiguous.

Page 3 line 14. Regarding "there may be conditions" please state what these conditions may be.

Page 4 line 6. Why were these mutations chosen to study in comparison to other mutations in this region of the protein?

Page 6 line 1. General comment on the Results in this manuscript:

In general, the quality of the work is high. However, I have reservations about the publication of manuscripts that are based solely on two biological replicates of recombinantly expressed protein. Most of the line-by-line critiques below reflect that concern. The paper uses four experiments (ATPase, single-turnover, motility, and single-molecule force spectroscopy) to draw conclusions. How many independent biological replicates of protein expression and purification were used for these experiment? Where there only two in total that were tested across all experiments? This should be stated more clearly in the methods and the number of biological replicates increased as stated in Public Review.

Page 7 line 7. How is the 16+/-9% being computed here? Is this from a single replicate or from the normalized data compared across the 2 biological replicates. Please state more clearly in the text how the effect size and error is computed. The reported effect sizes should be from multiple biological replicates.

Page 7 line 15. In Figure 2, why are the ATPase data displayed as representative ATPase curves? The data have been normalized and so the biological replicates should be able to be plotted on the same curve. Doing so increases transparency and quality of presentation. The normalized biological replicates should be able to be displayed on the same plot. Each replicate could be fit independently, or a single fit to the N = 2 (preferably N > 2) set of data fit to a single function.

Page 7 line 21. Please increase biological replicates used in these studies. In the N = 2 experiments noted here what is the +/- error being presented?

Page 8 line 3. Regarding "mean +/- SD." State what mean is being plotted and what SD is being plotted. Is this the mean of the three technical replicates from a single biological replicate or the mean of all data across multiple biological replicates. If it is biological replicates, SD is a rather limited error estimate as there are only two replicates.

Page 10 line 9. Are the single molecule experiments performed with biological replicates? Please do this if it wasn't done. The data do not make it clear which data sets in the single molecule experiments were from independent biological replicates. Please state how the reported mean values and SEM are computed in relationship to biological replicates. They should be computed across independent biological replicates as should statistical significance.

Page 11 Figure 3A. How many molecules are these curves determined from? Please include confidence intervals for these curves. Presumably, these plots should be determined from data across all biological replicates.

Page 11 Figure 3. As asked above, please explain which data are from a single biological replicate and which are from multiple replicates. The methods states that the single molecule experiments were performed from multiple biological replicates. Can the authors graphically indicate which molecules were from which prep? A table similar to S1, summarizing the statistics of the single molecule measurements would be highly valuable.

Page 13 Figure 4. Please provide confidence intervals with these curves and propagate the uncertainty from experimental measurements that they are calculated from, including uncertainty from biological replicates into the determination of these fundamental biophysical properties.

Page 14 Figure 6. As with Figure 2 ATPase data, please depict data of the control normalized biological replicates, not a single "representative" data set.

Page 15 Figure S2. Please provide additional detail on the experiment such as what replicate the data points come from. Do the data come from a single biological replicate or multiple biological replicates. Ideally, the data should be from more than 2 biological replicates.

Page 17 line 3. Is the N listed in this line and in Table S2, technical replicates or biological replicates? How many different preparations of protein contributed to this data?

Page 29 line 22-25. The authors state, "we have previously observed that additional biological replicates do not typically differ when the two biological replicates match" as justification for not providing additional replicates. Given the intrinsic uncertainty in all experiments performed in this study, please perform more biological replicates and ensure that all data presented in the paper are from more than two biological replicates of recombinant protein expression.

Page 30 line 17-21. Were the single turnover experiments performed using material obtained from more than one biological replicate? Please do so as indicated elsewhere in the review.

---

## [Author Response]

Essential revisions:Thank you for submitting your manuscript to eLife. It was examined by three expert reviews and myself. The general conclusion is that the work is thorough, significant and, for the most part, very well documented. All of the reviewers felt that if their concerns can be adequately addressed, the work merits publication.There were three major concerns. The first two deals with whether you have appropriate statistical power to back up your claims. One reviewer was concerned that the number of biological replicates is only two and this is magnified by the second concern which is the significant variability in the values reported in Table S1 for the WT sample. For example, the kcat varies from 4.64 to 6.04 /S in the two WT biological replicates and their Km values show even more variability for that replicate. Since all the data are normalized to the WT control, this raises concerns. In the manuscript you attribute this to "assay variation", but this degree of difference is surprising and reinforces the need for a third replicate. All of the reviewers appreciate the work involved in prepping and assaying six proteins and so we did not ask this lightly. Can you distinguish between assay variation and the case where the first WT replicate simply had less active motors? The data reported for the mutants are much tighter, but even their variation makes us wonder whether the small differences reported as deviations from WT are valid. Please either provide data from another replicate or give us strong reasons why we should not be concerned over this.

We appreciate the reviewers’ concerns with variability in the ATPase assays—we ourselves have previously noted this variability and understand why it may give pause. Our expression system allows us to express and highly purify human β-cardiac myosin fragments using mouse C2C12 myoblasts. Over the past ~11 years of using this system, we have observed variability in our measured k_cat_ values for both single- and double-headed WT myosin fragments (sS1, S1, 2hep, and 25hep) that we have sought to understand and minimize. We believe the single largest contribution to this variability is likely differences in post-translational modifications (PTMs) to myosin that occur during their production in the myoblast cell line. Indeed, we have performed mass spectrometry on selected purified samples and have seen that the list of modifications is extensive (>>100). While this finding is interesting, we have not pursued it both because of prohibitive cost and more importantly because such modifications, occurring in a heterologous expression system, are of unclear physiological relevance to the PTMs that occur in human cardiomyocytes. Please note that the most extensively studied PTM of the myosin motor, phosphorylation of the RLC, is controlled for in all our studies as we exchange on a bacterially expressed human RLC that is of course unphosphorylated. This stands in contrast to other groups who use this expression system but leave the endogenous mouse skeletal RLCs in place, with no control for phosphorylation status.

We hypothesize that alterations in PTMs may be due to subtle differences in cell density or differentiation state despite standardized protocols for cell growth and differentiation. These changes in turn may be secondary to slight differences in growth factors, etc. This issue has been exacerbated during the pandemic because of interruptions to lab access and supply chains. For example, the experiments reported in this manuscript were carried out over a period of 2-2.5 years, during which different batches of C2C12 cells (both from same ATCC stock), DMEM, fetal bovine serum and horse serum were necessarily used for myoblast cell culture.

Despite these issues, we have found over the course of studying over 50 mutations in human βcardiac myosin – that comparing proteins grown and harvested from the same batch of cells at the same time under the same conditions minimizes the effect of this variability and allows reproducible comparisons of the biochemical and biomechanical effects of any given mutant to its simultaneously prepared and assayed WT control. Indeed, we have seen remarkably consistent % changes from WT for a given mutation despite variations in absolute activity. Therefore, all of the data in this paper compares proteins that were expressed in parallel as described above. Please note that this means that the WT 2-*hep* to which the editor alludes was prepared independently on 12 different occasions (12 biological replicates) rather than two. We have also repeated the WT 2hep and WT 25hep protein preparations so that the updated data includes 5 biological replicates of this pair of proteins.

To address concerns around variability and number of replicates, we have done the following:

1. We have now provided three additional replicates of WT 2- vs. 25-*hep* to further validate the difference between WT 2- and 25-*hep*. These are reported in table S1 as replicates 3 – 5. When only two replicates were shown, the WT 25-*hep*/WT 2-*hep* ratio was 0.63 ± 0.10; with the addition of three biological replicates, the average ratio is now 0.61 ± 0.10. This additional data can be found in Figure 3 —figure supplement 1, and Supplementary File 1.

2. We include a new supplemental figure (Figure 3 —figure supplement 1) that shows all the nonnormalized ATPase replicates. Our goal in including the normalized data in the main text is to allow the reader to focus on the relevant differences between WT and mutants. Note that this normalization is to simultaneously prepared and assayed WT protein—we do not normalize to protein that was prepared on a different day or from a different batch of cells. The normalization eliminates slight background variability due to myoblast growth conditions, temperature, or protein age (as described above), allowing comparison of relevant differences between mutant and WT proteins. However, we recognize the value in including the raw data and have thus added Figure 3 —figure supplement 1.

We hope that these revisions help to allay the reviewers’ concerns about the variability in the ATPase results. The results that are used to make conclusions are statistically significant.

The third major concern deals with the optical trapping results. One reviewer commented that you should show some representative primary data showing single interactions and that a plot of displacement distributions would be important to see, given that these mutations are in the lever arm. More detail on the assumptions that go into the fittings of the harmonic force-spectroscopy data is also asked for.

We thank the reviewers for this valuable suggestion. We have now added a figure (Figure 4 —figure supplement 1) in the supporting information with representative primary data for each myosin assayed. The figure also includes plots of displacement distributions. We have also modified the text (page 22 line 23 to line 27) to better describe the assumptions that go into the fittings of harmonic force-spectroscopy data.

In addition, each reviewer had specific items that should be addressed.Reviewer #2 (Recommendations for the authors):P.22 Figure S3 Legend. Authors in discussing A797 stated "This region has previously been implicated in the formation of the folded state" and later on that F834 which is in the ankle joint that "this joint has been previously shown to be important in autoinhibition". I'm assuming that the folded state and autoinhibition are both referring to the IHM state. Also, no reference is given for these statements.

We have adjusted the text in the figure legend to refer more explicitly to the IHM state and have added references for these statements. We thank the reviewer for this comment.

Table S1 and P.7, line 5. The actin titrations of the myosin's ATPase activity aretypically fit to a rectangular hyperbola such as that used for analysis of the substrate dependence of enzyme kinetics with the Michaelis-Menten formalism. However, actin is not a substrate for myosin and so the term describing the actin concentration at half maximal Vmax should not be called a Km. Many authors use Kactin or Katpase to describe this value.

We have changed instances of “KM” to “Kapp” (as in, apparent K). We thank the reviewer for this comment.

Reviewer #3 (Recommendations for the authors):The authors need to better outline the assumptions that go into the fitting of the harmonic force-spectroscopy data. Would the analysis find non-linear mechanical behaviors of the lever caused by the mutation? What if the force-dependence didn't follow Bell's-Law like kinetics? I think the technique is totally appropriate for the study, but its limitations should be outlined.

We thank the reviewer for this valuable suggestion. Various aspects of the strengths and limitations associated with the harmonic force spectroscopy technique and assumption involved in the fitting of the harmonic force spectroscopy data has been described in detail previously by our research group (*Sung, J. et al.,* Methods in Enzymology, 2010, 475, 321-375; *Sun, J. et al.,* Nature Communications, 2015, 6, Article number 7931; *Liu, C. et al.,* Nature Structural Molecular Biology, 2018, 25, 505-514; and *Vander Roest, A. S. et al.,* Proceedings of the National Academy of the National Academy of Sciences U.S.A., 2021, 118, e2025030118.). However, we agree that a detailed explanation of certain strengths and weaknesses of the HFS method and corresponding data analysis would certainly improve the current manuscript and help the readers to perceive our results with greater insight. We have now modified the method section accordingly, to address this concern (page 22 line 23 to line 27).

What are the uncertainties on the plots in Figure 3A and Figure 4.

The uncertainty in Figure 3A (now Figure 4A) is now shown as dotted lines. We have also included the individual fits for each molecule, which is now shown in Figure 4 —figure supplement 1 M, N, O, P. The uncertainties for Figure 4 (now Figure 5) are also now shown as dotted lines. We thank the reviewer for this comment.

The Plots in Figures 2 and 5 are plotted as normalized values. Why not plot the ATPase values? It would be concerning if this is because there is too much variability from the assays.

We have now included full non-normalized values as Figure 3 —figure supplement 1. The purpose of showing normalized values in the main text is not to conceal any variability (as evidenced by the fact that we had already transparently included all of the individual fitted parameters for each experiment in table S1 – now Supplementary File 1), but to allow readers to focus on the relevant differences between mutant and WT.

It would be useful for Table S1 to report the mutant 25-Hep/WT 25-Hep kcat ratio.

Table S1 (now Supplementary File 1) is designed for readers to be able to see the real ATPase values for each independent biological replicate. As described in the methods, biological replicates of each mutant protein were only prepared with a paired WT 2-*hep* control, not a WT 25-*hep* control. Data for WT 25-*hep* was collected separately with its own WT 2-*hep* controls, as shown in the bottom row of the table. It would be misleading and inappropriate to normalize the mutant 25-*hep* values to the WT 25-*hep* values from separate, unpaired protein preparations. The rightmost column of Supplementary File 1 (average mutant 25-*hep*/mutant 2-*hep* ratios) is designed to allow the reader to make the relevant comparison.

Reviewer #4 (Recommendations for the authors):The critique below list specific questions for the authors to consider, ordered as they occur in the manuscript. Quotes indicate text in the manuscript on the indicated line.Page 2 line 8. Regarding "Frequent cause of hypertrophic…" can the authors state how frequent is frequent?

Based on a simple ClinVar search of HCM mutations, we found that 73 unique positions in the myosin head are associated with likely pathogenic or confirmed pathogenic HCM mutations out of a total 777 amino acids (~9%), while in the lever arm, 11 unique positions out of a total 61 amino acids (~18%) in the lever arm are associated with likely pathogenic or confirmed pathogenic HCM mutations, suggesting ~2x enrichment for mutations in the lever arm. However, we know that the ClinVar database does not include all known HCM mutations, and that the levels of evidence vary somewhat from submitter to submitter, so this is a relatively weak analysis that we do not feel is adequate for publication. Thus, we have relied on the word “frequent” to convey this sentiment. We believe a separate bioinformatic analysis would be warranted to determine how much more frequent these mutations are, but we believe that analysis is outside the scope of the current work.

Page 2 line 12. Please avoid ambiguous and qualitative terms such as "relatively high" and "frequent" (see previous comment) as they are… ambiguous.

See comment above. While we appreciate the importance of avoiding ambiguity, an inherent challenge of understanding the frequency of mutations in a human disease is the lack of available high-quality data that exhaustively annotates every known mutation and whether the mutation is benign, potentially pathogenic, or definitively pathogenic. However, our rudimentary analysis suggests that “relatively high” and “frequent” are warranted. We are not aware of any publication that rigorously assesses the frequency of HCM-causing lever arm mutations.

Page 3 line 14. Regarding "there may be conditions" please state what these conditions may be.

We have added the parenthetical “for example, in the presence of drugs,” which is supported by reference 35, where they showed a difference in the relative increase in IHM vs. SRX upon the addition of mavacamten. Anecdotally, we have also noted that omecamtiv mecarbil may uncouple the relationship between the structural state and SRX-like rates (unpublished data).

The phrase is intentionally ambiguous because the goal is to point out that an individual rate constant can arise from multiple protein states; an individual rate is, by rule, not necessarily definitive proof of a specific structural state. In this case, there’s no reason to believe that an SRXlike rate couldn’t arise from any number of different structural states myosin may take. This point is discussed at-length in our prior publication (see ref. 24).

Page 4 line 6. Why were these mutations chosen to study in comparison to other mutations in this region of the protein?

We have added justification for selecting these mutations (page 4 lines 7-10), and we thank the reviewer for this comment.

Page 6 line 1. General comment on the Results in this manuscript:In general, the quality of the work is high. However, I have reservations about the publication of manuscripts that are based solely on two biological replicates of recombinantly expressed protein. Most of the line-by-line critiques below reflect that concern. The paper uses four experiments (ATPase, single-turnover, motility, and single-molecule force spectroscopy) to draw conclusions. How many independent biological replicates of protein expression and purification were used for these experiment? Where there only two in total that were tested across all experiments? This should be stated more clearly in the methods and the number of biological replicates increased as stated in Public Review.

We hope that the analysis and adjustments reported in the “essential revisions” section will help allay the reviewer’s concern in this regard. The results reported in this paper represent a great number of individual biological replicates across all samples and mutants.

Page 7 line 7. How is the 16+/-9% being computed here? Is this from a single replicate or from the normalized data compared across the 2 biological replicates. Please state more clearly in the text how the effect size and error is computed. The reported effect sizes should be from multiple biological replicates.

16 ± 9% is a composite of the two biological replicates. To compute the error, the errors of the fit for each biological replicate (17 ± 11% and 15 ± 7%) were propagated using the standard formula for error propagation in a mean:Δχest=Δχ21+Δx22N Error propagation was used instead of SD due to the limitation of only having two biological replicates, which makes SD inappropriate. Note that error propagation resulted in larger errors as compared to computed standard deviations for the biological replicates (which is appropriate in this context). We have clarified our use of error propagation in the corresponding methods section.

Page 7 line 15. In Figure 2, why are the ATPase data displayed as representative ATPase curves? The data have been normalized and so the biological replicates should be able to be plotted on the same curve. Doing so increases transparency and quality of presentation. The normalized biological replicates should be able to be displayed on the same plot. Each replicate could be fit independently, or a single fit to the N = 2 (preferably N > 2) set of data fit to a single function.

Due to edits suggested in the “essential revisions” section, we have now included Figure 3 —figure supplement 1, which shows all of the non-normalized replicates for all of the reported actin-activated ATPase results. We hope this will adequately address the reviewer’s concern around transparency.

From a technical perspective, plotting the individual replicates on the same chart results in a different fitting result as compared to plotting the replicates separately and averaging the resulting fitting parameters (even when the experiments are both normalized). While there are numerous mathematical reasons for this difference, a key technical reason in the context of these experiments is that the KM’s vary across biological replicates. Additionally, increasing the number of datapoints in a fitted function artificially reduces the standard error of the fit, making composite data appear misleadingly certain. For these reasons, we do not think it would be appropriate to fit data from separate biological replicates to a single function.

Page 7 line 21. Please increase biological replicates used in these studies. In the N = 2 experiments noted here what is the +/- error being presented?

This is the same data described in page 7 line 7 above, which we address in the response to the comment above. We recognize that the increase of 16% is marginal, but it is statistically significant and therefore relevant to report (though statistical significance is not necessarily indicative of biological significance). In fact, we would have preferred if this increase was *not* significant, as it’s surprising on its own that a mutation in the pliant region, fairly distal from the active site, could have an effect on the ATPase rate. Nevertheless, when differences are statistically significant, we have an obligation to report them.

The conclusions in the paper on the function of D778V do not hinge on this individual increase of 16%*,* but instead take into account all of the observed effects of the D778V mutation on motor function, including findings from the optical trapping and in vitro motility. For example, in figure 4C (overall power output), if the 16% increase in ATPase is not used to compute power output and instead we use an equivalent ATPase to the WT, the power output for D778V is still greater than that of WT. The increased power output is primarily due to the increased velocity of D778V which is, in turn, due to its increased detachment rate observed in single molecule optical trapping. The increased velocity of D778V as measured by the function of detachment rate and step size (equation 5) is corroborated by the observed increase in in vitro motility velocity.

Page 8 line 3. Regarding "mean +/- SD." State what mean is being plotted and what SD is being plotted. Is this the mean of the three technical replicates from a single biological replicate or the mean of all data across multiple biological replicates. If it is biological replicates, SD is a rather limited error estimate as there are only two replicates.

Standard deviations here represent the standard deviation of the data points shown in the chart, which collectively come from at least 3 separate protein preparations for each protein (as noted above in Author response table 1 in the response to the question on page 6 line 1). This has been clarified in the figure legend.

**Author response table 1. sa2table1:** Full numbers of biological and technical replicate.

Assay	Protein	Biological Replicates	Technical Replicates
Light-chain loadingassay(Each biological replicate is from a single unique protein prep diluted across 5 gel lanes, implying 5 technical replicates per biological replicate)	WT 2-*hep*	9	45
	D778V 2-*hep*	3	15
	L781P 2-*hep*	3	15
	S782N 2-*hep*	3	15
	A797T 2-*hep*	3	15
	F834L 2-*hep*	3	15
Actin-activatedATPase assay(Biological replicates are from separate protein preps, technical replicates are from separate wells on the same plate)	WT 2-*hep*	15	45
	WT 25-*hep*	5	15
	D778V 2-*hep*	2	6
	D778V 25-*hep*	2	6
	L781P 2-*hep*	2	6
	L781P 25-*hep*	2	6
	S782N 2-*hep*	2	6
	S782N 25-*hep*	2	6
	A797T 2-*hep*	2	6

In the context of the in vitro motility experiments (and optical trapping experiments), the definitions of technical and biological replicates are not necessarily immediately evident. For example, for each slide prepared, three separate imaging frames are averaged together to collectively determine the MVEL of a given slide channel. In that sense, each data point shown is already a summary of triplicate measurements. Additionally, some people would consider aliquots of the same protein prep but thawed and assayed on separate days using new stocks of actin and other motility proteins as separate biological replicates. In light of this complexity, it makes most sense to treat each slide as an individual replicate, which is what we have done here. As mentioned above, each slide is measured in triplicate.

Page 10 line 9. Are the single molecule experiments performed with biological replicates? Please do this if it wasn't done. The data do not make it clear which data sets in the single molecule experiments were from independent biological replicates. Please state how the reported mean values and SEM are computed in relationship to biological replicates. They should be computed across independent biological replicates as should statistical significance.

Single molecule experiments were performed across 2-3 independent protein preparations, as noted in the table in the response to the reviewer’s question above (page 6 line 1). We have also added Figure 4 —figure supplement 2, which shows the data points colored to indicate which independent protein preparation the molecules originated from. Molecules from unique protein preparations do not appear to systematically differ from molecules from separate protein preparations.

In many ways, the optical trapping experiments present a similar complication in the definition of technical and biological replicates as the in vitro motility experiments, discussed above in response to the question for page 8 line 3. For each molecule, multiple independent binding events are collected (see newly included figures Figure 4 —figure supplement 1 E-H, where each data point represents an individual binding event from an individual molecule). These measurements are summarized into a force-velocity fit for each molecule (Figure 4 —figure supplement 1 I-L), which represents summary data from many individual measurements. Therefore, it makes most sense to treat each molecule as an independent replicate, which is what we have done here. While error bars for each molecule are not shown to preserve visual clarity, each molecule does have its own error measurement for *k*_0_ and δ. (Step size *d* is calculated as an averaged summary of all of the events from an individual molecule; thus each molecule does not have a measured error in step size.)

Page 11 Figure 3A. How many molecules are these curves determined from? Please include confidence intervals for these curves. Presumably, these plots should be determined from data across all biological replicates.

The curves in Figure 3A show the average fit across all molecules. The number of molecules for each protein has been added to the figure legend, and we have now shown the error of the averages as dotted lines in Figure 3A. The fits for each individual molecule are now also shown in Figure 4 —figure supplement 1 M-P.

Page 11 Figure 3. As asked above, please explain which data are from a single biological replicate and which are from multiple replicates. The methods states that the single molecule experiments were performed from multiple biological replicates. Can the authors graphically indicate which molecules were from which prep? A table similar to S1, summarizing the statistics of the single molecule measurements would be highly valuable.

We have added Figure 4 —figure supplement 2, which shows the data points colored to indicate which independent protein preparation the molecules originated from. We thank the reviewer for this comment.

Page 13 Figure 4. Please provide confidence intervals with these curves and propagate the uncertainty from experimental measurements that they are calculated from, including uncertainty from biological replicates into the determination of these fundamental biophysical properties.

Figure 4 (now Figure 5) now shows confidence intervals for each curve, which are propagated from the uncertainty of experimental measurements of each protein. We thank the reviewer for this comment.

Page 14 Figure 6. As with Figure 2 ATPase data, please depict data of the control normalized biological replicates, not a single "representative" data set.

We have replaced the main text versions of these figures with plots showing the fitted curves for each kinetic assay, along with an average fitted curve for each WT and mutant 25-*hep*. We have also added the same plots for the supplemental figure showing WT and mutant 2-*hep* single turnover results (now Figure 7 —figure supplement 2), and moved the representative fitted curves to a new supplemental figure (now Figure 7 —figure supplement 1). We thank the reviewer for this comment.

Page 15 Figure S2. Please provide additional detail on the experiment such as what replicate the data points come from. Do the data come from a single biological replicate or multiple biological replicates. Ideally, the data should be from more than 2 biological replicates.

We hope the table shown in our response to the reviewer’s earlier point (page 6 line 1) helps clarify biological and technical replicates for these experiments. We have also now noted the number of unique protein preparations for each protein construct in the corresponding methods section. We thank the reviewer for this comment.

Page 17 line 3. Is the N listed in this line and in Table S2, technical replicates or biological replicates? How many different preparations of protein contributed to this data?

As above, we hope Author response table 1 in our response to the reviewer’s earlier point (page 6 line 1) helps clarify, along with the adjustment to the methods section.

Page 29 line 22-25. The authors state, "we have previously observed that additional biological replicates do not typically differ when the two biological replicates match" as justification for not providing additional replicates. Given the intrinsic uncertainty in all experiments performed in this study, please perform more biological replicates and ensure that all data presented in the paper are from more than two biological replicates of recombinant protein expression.

We hope that our response to the “essential revisions” has helped allay the reviewer’s concern in this regard.

Page 30 line 17-21. Were the single turnover experiments performed using material obtained from more than one biological replicate? Please do so as indicated elsewhere in the review.

As above, we hope Author response table 1 in response to the reviewer’s earlier point (page 6 line 1) helps clarify, along with the adjustment to the methods section.